# Grid Refinement in ICON v2.6.4

Günther Zängl[1], Daniel Reinert[1], and Florian Prill[1]

[1]Deutscher Wetterdienst, Offenbach am Main, Germany

**Correspondence:** Günther Zängl (Guenther.Zaengl@dwd.de)

**Abstract.** This article describes the implementation of grid refinement in the ICOsahedral Nonhydrostatic (ICON) modeling system. It basically follows the classical two-way-nesting approach known from widely used mesoscale models like MM5 or WRF, but differs in the way how feedback from fine grids to coarser grids is applied. Moreover, the ICON implementation supports vertical nesting in the sense that the upper boundary of a nested domain may be lower than that of its parent domain.

Compared to the well-established implementations on quadrilateral grids, new methods had to be developed for interpolating the lateral boundary conditions from the parent domain to the child domain(s). These are based on radial basis functions (RBFs) and partly apply direct reconstruction of the prognostic variables at the required grid points, whereas gradient-based extrapolation from parent to child grid points is used in other cases. The run-time flow control is written such that limited-area domains can be processed identically to nested domains except for the lateral boundary data supply. To demonstrate the func-

tionality and quality of the grid nesting in ICON, idealized tests based on the Jablonowski-Williamson test case (Jablonowski and Williamson, 2006) and the Schär mountain wave test case (Schär et al., 2002) are presented. The results show that the numerical disturbances induced at the nest boundaries are small enough to be negligible for real applications. This is confirmed by experiments closely following the configuration used for operational numerical weather prediction at DWD, which demonstrate that a regional refinement over Europe has a significant positive impact on the forecast quality in the northern

hemisphere.

## 1 Introduction

The ICON (ICOsahedral Nonhydrostatic) modeling framework is jointly developed by the German Weather Service (DWD), the Max Planck Institute for Meteorology (MPI-M), the German Climate Computing Center (DKRZ) and the Karlsruhe Institute for Technology (KIT), targeting a unified global numerical weather prediction (NWP) and climate modeling system

(GCM). The development work started in 2004 with basic research on the model grid and the numerical formulation of the dynamical core in a highly idealized shallow-water framework (Bonaventura and Ringler, 2005; Rípodas et al., 2009). The next step was the implementation of a hydrostatic dynamical core using the same model equations, time-integration scheme, and vertical discretization as the spectral transform ECHAM model (Wan et al., 2013). This work focused on investigating the effects of the different horizontal grid formulation (icosahedral vs. spherical harmonics) and provided the basis for a first

version with full physics coupling. In parallel, nonhydrostatic dynamical cores were developed on hexagonal grids (Gassmann and Herzog, 2008; Gassmann, 2013) and triangular grids (Zängl et al., 2015). As discussed in Gassmann (2011) and Danilov

(2012), C-grid type triangular discretizations suffer from a spurious computational mode, which manifests itself in a rapidly oscillating checkerboard pattern in the horizontal divergence field, giving rise to numerical stability problems. To date, this problem is lacking a rigorous mathematical solution. However, Zängl et al. (2015) showed that the problem could largely be mitigated by a specific averaging of the velocity components entering the divergence operator. This pragmatic solution allowed to retain the triangular C-grid discretization. Moreover, a triangular grid proved to be more suitable for implementing a two-way grid nesting capability than a hexagonal one. Triangular cells can be recursively partitioned into successively smaller triangles, which leads to a unique relationship between parent and child cells. For hexagons, however, this is not the case, as the majority of child cells is shared between two adjacent parent cells. To our knowledge, this is one aspect that makes two-way grid refinement approaches developed for hexagonal grids significantly more complex (e.g. Dubos and Kevlahan, 2013).

Despite impressive advances in computational power over the last decades, the application of global models with uniform, convection-permitting resolution on weather or even climate timescales is still too costly to be performed on a regular basis. To date, high-resolution limited-area models (LAM) serve as a cost-effective alternative for exploring the meso- and microscale, and will continue to serve as a working horse for both the NWP and climate community. Limited-area models have proven successful, but they are known to have conceptional deficiencies, such as the potential ill-posedness of lateral boundary conditions (Davies, 2014), possible inconsistencies with the driving model in terms of the governing equations, numerical formulations or physical parameterizations, and the lack of regional-to-global scale interactions (Warner et al., 1997). In order to mitigate these deficiencies, a number of methods have been considered to achieve locally enhanced resolution in global models.

The oldest approach dating back more than 40 years is the usage of stretched grids (Schmidt, 1977; Staniforth and Mitchell, 1978). The so-called Schmidt transformation allows for enhanced resolution in a particular region of interest by redistributing the grid points of an initially uniform model grid. This approach has proven successful in various NWP and climate applications. A review on GCM applications is provided, e.g. by Fox-Rabinovitz et al. (2008), and Goto et al. (2015) discuss a more recent NWP application using NICAM, which is based on a modified Schmidt transformation on an icosahedral grid (Tomita, 2008). Grid stretching obviates the need for lateral boundary conditions, and it allows for an immediate interaction between global and regional scales. Probably the largest drawback of this method is the fact that grid points can only be redistributed rather than created. Enhancing the resolution in one region of the globe implies a coarsening in another region. Besides the inevitable coarsening itself, the insufficient resolution of disturbances passing through the coarsened region may also negatively affect the simulation in the refined region.

More recently, global models using locally refined unstructured meshes have been developed. Unlike the grid stretching approach, they allow new grid cells to be added in regions of interest (static h-refinement). This approach is pursued, for example, by the Model for Prediction Across Scales (MPAS) (Skamarock et al., 2012), or the spectral element dynamical core of the Community Atmosphere Model (CAM) (Zarzycki et al., 2014). Albeit being much more flexible, this approach faces similar challenges as the grid stretching approach mentioned previously. In both approaches, the time step is restricted by the smallest cell in the domain, unless specific measures like substepping are taken for individual cells or regions, or horizontally implicit numerical methods are chosen. Moreover, care must be taken that the parameterizations are applicable on a wide range of scales, and turn off gradually when the respective processes become resolved on the model grid (such as the convection

parameterization in the gray zone). The development of scale aware parameterizations poses major challenges and is an area of active research (Gross et al., 2018). However, notable progress in terms of scale awareness has been reported e.g. for the CAM5 parameterization suite, when compared to CAM4 (Gettelman et al., 2018).

The approach we are pursuing closely resembles traditional two-way nesting, as known from many regional mesoscale models such as MM5 (Grell et al., 1994) or WRF (Skamarock et al., 2019). Two-way nesting differs from the previous single-grid approaches by the fact that multiple grids of different resolution are overlaid onto each other, such that individual points on the globe are covered by more than one prognostic grid cell. Scale interaction can be ensured by feeding the nested-grid solution back to the underlying parent grid at regular time intervals. Similar to LAMs, this method requires lateral boundary

conditions to be specified for every nested domain, and it may suffer from spurious wave reflections at nest boundaries due to the abrupt jump in resolution. On the other hand, it allows many model settings to be chosen individually for each domain, which improves numerical efficiency and reduces the requirements concerning the parameterization suite. Distinct time steps can be chosen for each domain, allowing e.g. a proportional reduction in refined domains in order to meet stability constraints. The parameterizations can be tuned individually for each domain and resolution, or even switched off, relaxing the requirement

of scale awareness to some degree. While two-way nesting has been successfully applied in limited-area modeling since decades, it is still much less common in global modeling. Focusing on recent global models which are being used in research or operational forecasting, the authors are only aware of the two-way nesting approach implemented in the Geophysical Fluid Dynamics Laboratory (GFDL)'s Finite-Volume Cubed-Sphere Dynamical Core (FV3) (Harris and Lin, 2013, 2014; Mouallem et al., 2022).

The purpose of this article is to describe the implementation of grid nesting in ICON. Compared to previous two-way-nesting approaches, it is the first implementation on triangular grids and differs in the way how feedback from fine grids to coarser grids is realized. In addition, the ICON implementation allows for vertical nesting in the sense that the upper boundary of a nested domain may be lower than that of the parent domain, which is not supported by most previous nesting implementations in other models. Without specific discussion, we note that a limited-area mode is available in ICON as a by-product of the grid

nesting implementation, differing from nesting only in the way how the lateral boundary conditions are provided. A detailed description the grid nesting implementation will be provided in Sect. 2, followed by application examples in Sect. 3. A brief summary will be given in Sect. 4.

## 2    Domain Nesting in ICON[1]

### 2.1    Description of the Basic Design

The understanding of ICON's nesting implementation requires some basic knowledge of ICON's mathematical-physical design. We therefore start by summarizing key elements of the dynamical core, time stepping and physics-dynamics coupling scheme. ICON's dynamical core solves the fully compressible, nonhydrostatic Euler equations on the sphere, using either the

---

[1]Some elements of this description have already been published in the ICON User tutorial (Prill et al., 2020).

shallow or deep atmosphere formulation (Borchert et al., 2019). The spatial discretization is performed on an unstructured icosahedral-triangular Arakawa-C grid in the horizontal, and a terrain following height-based SLEVE coordinate (Leuenberger et al., 2010) with Lorenz-type staggering in the vertical. As described in Zängl et al. (2015), the prognostic variables encompass the edge-normal horizontal wind speed $v_n$, vertical wind speed $w$, total air density $\rho$, virtual potential temperature $\theta_v$, and mass fractions $q_k$ of various moisture quantities. See Fig. 1 in Wan et al. (2013) for the variable placement on the triangular grid. The two-time-level predictor-corrector time integration scheme is fully explicit except for the terms describing the vertical propagation of sound waves, which are treated implicitly (Zängl et al., 2015). Hence, the permitted integration time step is comparatively small and constrained by the ratio of the speed of sound to the horizontal mesh size.

To optimize computational efficiency, different integration time steps are used for the dynamical core on the one hand and additional sub-grid physical processes (and tracer transport) on the other hand. The time steps will be denoted by $\Delta\tau$ for the dynamical core and $\Delta t$ for physics. The dynamical core is sub-stepped with respect to physical processes, with a single physics time step consisting of 5 dynamics substeps by default ($\mathtt{nsubs} = \Delta t/\Delta\tau = 5$).

The physics-dynamics coupling scheme in ICON further distinguishes between *fast* physical processes, having a time scale shorter than or comparable to the time step $\Delta t$, and *slow* processes, which are considered to have a time scale large compared to $\Delta t$. Processes that fall into the category *fast* are saturation adjustment, grid scale microphysics or turbulent diffusion, while examples of *slow* processes are convection and radiation. Fast processes are integrated with the previously defined time step $\Delta t$, whereas slow processes may be integrated with process-specific larger time steps that are an integer multiple of $\Delta t$. We therefore call the time step $\Delta t$ the fast physics time step. As the tracer transport is performed at $\Delta t$ as well, we apply the coupling of nested domains at $\Delta t$ for both dynamics and tracer variables because this simplifies maintaining consistency with continuity (Gross et al., 2002). Essentially it is required to provide time-averaged mass fluxes (averaged over the substeps) for tracer transport at nest lateral boundaries (see Sect. 2.2.1), and to ensure that the child-to-parent feedback increments for partial densities $\rho q_i$ sum up to the feedback increments for total density $\rho$ (see Sect. 2.2.2).

The static mesh refinement in ICON is accomplished using multiple individual grids, where one or more higher resolution (child) domains are overlaid on a coarser base (parent) domain. The base domain can be a regional or a global domain. Model integration on the child domain is performed in addition to that on the underlying part of the parent domain, i.e. there is no 'hole' in the parent domain where the child domain is located.

Each child domain has a defined parent domain providing lateral boundary conditions, but a parent domain can have several child domains. The child domains can be located in different geographical regions and can also serve as parent domains for further subdomains, but domains having the same parent are not allowed to share the same parent grid cells because this would lead to ambiguities in combination with two-way nesting. Nested domains may also be switched on or off during runtime. Conceptually, the number of nested domains is arbitrary and controlled by the grid files provided as input, but of course not all choices would make sense from a physical point of view. Each domain can be regarded as separate instances of the same model that are coupled to each other, using the same numerical operators and filters, time integration scheme and physics-dynamics coupling. If desired, however, different physical settings can be chosen individually for each domain. For example,

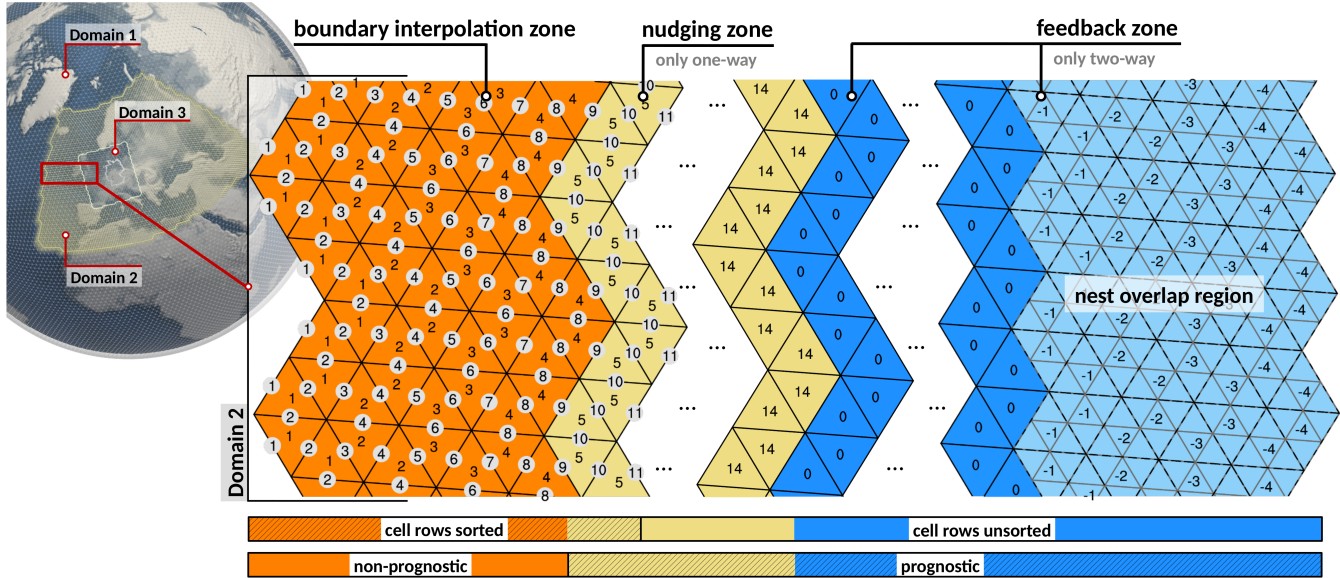

**Figure 1.** Basic structure of a nested (or limited-area) domain, exemplified by a section of domain 2 shown in the upper left. Orange: boundary interpolation zone, having a fixed width of 4 cell rows. Ocher: nudging zone with adjustable width, only active for one-way nesting and in limited-area mode. Blue and light-blue: child-to-parent feedback zone. Light-blue: nest overlap region, i.e. a region for which a higher resolution child domain exists (see domain 3 in the schematic on the upper left). Prognostic computations are restricted to the feedback and nudging zone. Integers indicate the internal indexing of domain 2 which is used to assign cells and edges to individual zones. More details on the indexing and the indicated sorting of cell rows are given in the Appendices A1 and A2.

radiation can be called more frequently on subdomains, or the convection scheme can be tuned differently or even be switched off completely.

The refinement ratio between the parent domain and a child domain is fixed to a value of 2, as each parent triangle is split 130 into 4 child triangles (see the right part of Fig. 1). While higher refinement ratios would technically be possible, we decided for this restriction because it reduces the risk for numerical artifacts (e.g. by partial wave reflection) along nest boundaries. Consistent with the refinement ratio of 2, the model integration time step $\Delta t$ is multiplied by a factor of $0.5$ for each additional nesting level. The coupling time step between successive nesting levels is the fast physics time step $\Delta t$.

The parent-child coupling can be either one-way or two-way. A mixture of one-way and two-way coupled domains is also 135 possible. Two-way versus one-way coupling means that the prognostic variables on the child domain are transferred back to the coarser parent domain at regular time intervals using a dedicated feedback mechanism. As a result, the solution on the parent domain benefits from the higher resolution of the child domain. In case of one-way nesting, the feedback is switched off.

To perform the coupling, we conceptionally split any nested domain into three zones, which we call the *boundary interpolation zone*, the *nudging zone* and the *feedback zone*. In order to identify grid points belonging to these zones, cells, edges 140 and vertices are indexed according to their distance from the boundary (see Appendix A1 for details). These zones along with

the grid point indexing are depicted in Fig. 1, which shows part of the boundary region and inner region of a nested domain. Limited-area domains are treated technically like one-way nested domains except for the fact that the boundary interpolation zone is filled with external input data rather than data interpolated from the parent domain.

The *boundary interpolation zone* is non-prognostic, and is meant to store the boundary conditions that are necessary to solve the governing equations in the child domain. Boundary conditions are needed for the prognostic variables $v_n$, $w$, $\rho$, $\theta_v$, and $q_k$. By a dedicated boundary update mechanism (see Sect. 2.2.1 for details), both the prognostic variables and their time tendencies are interpolated from the parent to the child domain, and the boundary conditions are updated at every child time step. The boundary interpolation zone has a fixed width of 4 cell rows. This is motivated by the technical constraint that the boundary zone needs to match with parent cell rows (i.e. an odd number of cell rows is not allowed), combined with the fact that 2 cell rows would not be sufficient to cover all stencil operations performed in the dynamical core. For example, the $\nabla^4$-diffusion operator (Zängl et al., 2015) requires information from three adjacent cell rows. We note that halving the size of the boundary interpolation zone would be possible by modifying some stencil operations near the lateral boundary, such as replacing $\nabla^4$ with $\nabla^2$. However, this would have no impact on computational efficiency in practical applications with MPI domain decomposition (see Appendix A2).

The *nudging zone*, which is active in the case of one-way nesting only, serves to damp differences between the driving solution in the adjacent boundary interpolation zone and the prognostic solution in the child domain. Essentially, the prognostic model state of the child domain is relaxed (nudged) to the parent state, following the traditional Newtonian-relaxation approach described by Davies (1976). Details of the implementation are provided in Sect. 2.2.3. For two-way nesting, no nudging is applied and the boundary interpolation zone borders on the feedback zone.

In the *feedback zone*, the model state on the parent domain is relaxed towards the updated model state on the child domain at every fast physics time step $\Delta t_p$ of the parent domain. We refer to this as *relaxation-type* feedback. As a result, the parent and child domains remain tightly coupled, and the solution on the parent domain benefits from the enhanced resolution of the child domain. Feedback is applied to the prognostic variables $v_n$, $w$, $\rho$, $\theta_v$ as well as to the mass fractions of water vapour $q_v$, cloud water $q_c$, and cloud ice $q_i$. Child-to-parent feedback has already been successfully applied in mesoscale models like MM5 (Grell et al., 1994) or WRF (Skamarock et al., 2019), as well as in global simulations based on a cubed-sphere grid (Harris and Lin, 2013). However, our approach differs in the sense that the parent state is relaxed towards the child state with an adjustable timescale, rather than being overwritten by the child state. Compared to the conventional direct feedback, which is available as an option in ICON as well, the relaxation feedback has the advantage of generating less numerical disturbances near vertical nest interfaces, leading to slightly better forecast quality in NWP applications (without vertical nesting, the quality difference is small). In addition, the relaxation feedback requires no adjustment of the model orography at the parent grid level, which is more convenient if nested domains are turned on or off during runtime. See Sect. 2.2.2 for further details.

## 2.2 Parent-Child Coupling

### 2.2.1 Lateral Boundary Update: Parent → Child

The boundary update mechanism provides the child domain with up-to-date lateral boundary conditions for the prognostic variables $v_n$, $w$, $\rho$, $\theta_v$, $q_k$. In order to prevent parent-to-child interpolated values of $\rho$ from entering the solution of the mass continuity equation, the above set of variables is extended by the horizontal mass flux $\rho v_n$. This will allow for parent-child mass flux consistency, as described below. For the subsequent description of the algorithm, let the model state on the parent and child domain be denoted by $\mathcal{M}_p^n$ and $\mathcal{M}_c^n$, respectively, where $n$ specifies the time step index.

In general, the boundary update works as follows: Let $\psi_p^n$, $\psi_p^{n+1}$ represent a prognostic variable on the parent domain at time steps $n$ and $n+1$, respectively. Once the variables on the parent domain $\mathcal{M}_p$ have been updated from $n$ to $n+1$, the time tendency

$$\frac{\partial \psi_p}{\partial t} = \frac{\psi_p^{n+1} - \psi_p^n}{\Delta t_p}$$

is diagnosed. Both the field $\psi_p^n$ at time level $n$ and the tendency $\frac{\partial \psi_p}{\partial t}$ are then interpolated (downscaled) from the parent grid cells/edges to the corresponding cells/edges of the child domain's boundary zone (orange cells in Fig. 1). With $\mathcal{I}_{p \to c}$ denoting the interpolation operator, we get

$$\psi_c^n = \mathcal{I}_{p \to c}\left(\psi_p^n\right)$$
$$\frac{\partial \psi_c}{\partial t} = \mathcal{I}_{p \to c}\left(\frac{\partial \psi_p}{\partial t}\right).$$

The interpolated tendencies are needed in order to provide the lateral boundary conditions at the right time levels, since two integration steps are necessary on the child domain in order to reach the model state $\mathcal{M}_c^{n+1}$, with each step consisting of several (typically 5) dynamics sub-steps. The temporal update is performed at each dynamics sub-step $\Delta \tau_c$ for the prognostic variables of the dynamical core and at each large step $\Delta t_c$ for the tracer variables. As an example, the boundary conditions at the first dynamics sub-step of the first and second (fast physics) integration step on the child domain read $\psi_c^n$ and $\psi_c^n + 0.5\,\Delta t_p\,\partial \psi_c/\partial t$, respectively.

Concerning the interpolation operator $\mathcal{I}_{p \to c}$, we distinguish between cell-based variables (i.e. scalars) and edge-based variables ($v_n$ and $\rho v_n$). For cell-based variables, a 2D horizontal gradient is reconstructed at the parent cell circumcenter by first computing edge-normal gradients at edge midpoints, followed by a 9-point reconstruction of the 2D gradient at the cell center based on radial basis functions (Narcowich and Ward, 1994). The interpolated value at the $j$th child cell center is then calculated as

$$\psi_{c_j} = \psi_p + \nabla \psi_p \cdot \boldsymbol{d}(p, c_j), \qquad \forall j \in \{1 \ldots 4\}, \tag{1}$$

with $\nabla \psi_p$ denoting the horizontal gradient at the parent cell center, and $\boldsymbol{d}(p, c_j)$ the distance vector between the parent and $j$th child cell center. The same operator is applied to cell-based tendencies.

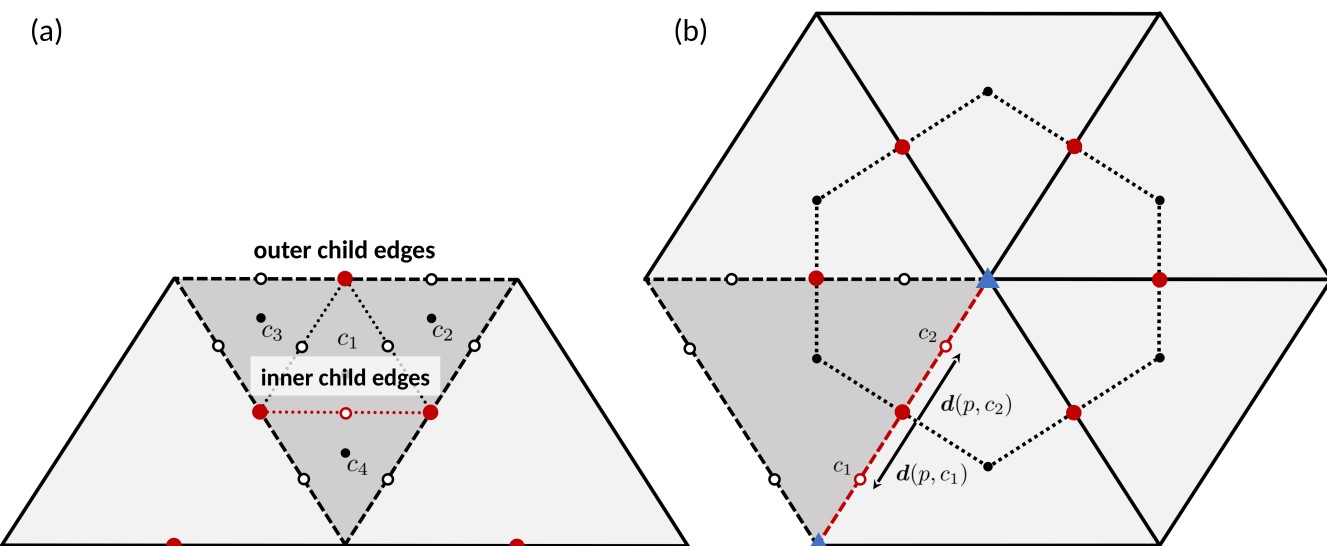

**Figure 2.** Horizontal reconstruction stencil for edge-normal vector components at (a) inner child edges and (b) outer child edges. The child edge under consideration is highlighted in red. Black open dots indicate child edge midpoints, while black solid dots indicate cell circumcenters. Solid red dots represent the reconstruction stencil, i.e. the location of the parent edge-normal vector components entering the reconstruction, and blue triangles in (b) indicate the location of the reconstructed 2D vectors. See the text for details.

To prevent excessive over- and undershoots of $\psi_{c_j}$ in the vicinity of strong gradients, a limiter for $\nabla\psi_p$ is implemented. It ensures that

$$\frac{1}{\beta}\psi_{p,\min} < \psi_{c_j} < \beta\psi_{p,\max} \qquad \forall j \in \{1\ldots4\}$$

on all four child points, where $\psi_{p,\min}$ and $\psi_{p,\max}$ denote the minimum and maximum of $\psi_p$, respectively, on the above-mentioned reconstruction stencil including the local cell center, and $\beta = 1.05$ is a tuning parameter. To minimize interpolation errors above steep orography, perturbations from the reference state (Zängl et al., 2015) rather than the full values are interpolated for the thermodynamic variables $\rho$ and $\theta_v$. In addition, the interpolation operator $\mathcal{I}_{p\to c}$ is also applied to the model orography in the boundary interpolation zone in the setup phase of the model (before calculating the vertical grid) in order

to ensure consistency of the model grids even if different raw data sets have been used to generate the model orographies. To allow a smooth transition to the orography in the interior of the nested domain, a linear blending with a width of eight cell rows (as for the above-mentioned nudging zone) is applied.

Regarding the interpolation of edge-based variables (i.e. the edge-normal vector components $v_n$ and $\rho v_n$), we distinguish between *outer child edges* coinciding with the edges of the parent cell, and *inner child edges* (see Fig. 2a).

Edge-normal vector components at the inner child edges are reconstructed using a direct RBF reconstruction based upon the five-point stencil indicated by solid red dots in Fig. 2a. For a given inner child edge the stencil comprises the edges of the

corresponding parent cell, and the two edges of the neighboring parent cells that (approximately) share the orientation of the inner child edge.

For the outer child edges, a more sophisticated reconstruction is applied in order to ensure that the mass flux across a parent edge equals the sum of the mass fluxes across the corresponding child edges. We start with an RBF reconstruction of the 2D vector of the respective variable at the parent triangle vertices (blue triangles in Fig. 2b), using the six (five at the original vertices of the icosahedron, the *pentagon points*) edge points adjacent to a vertex (red dots).

The edge-normal vector component $\phi$ at the child edge is then computed as

$$\phi_{c_e} = \phi_p + \nabla_t \phi_p \cdot \boldsymbol{d}(p, c_e), \qquad \forall e \in \{1, 2\}, \tag{2}$$

with $\boldsymbol{d}(p, c_e)$ denoting the distance vector between the parent and child edge midpoints for a given parent edge, and $\nabla_t \phi_p$ denoting the gradient of the edge-normal vector component $\phi_p$ tangential to the parent edge. The latter is computed by projecting the reconstructed 2D vectors at the two vertices of an edge onto the edge-normal direction and taking the centered difference. Since by construction $\boldsymbol{d}(p, c_1) = -\boldsymbol{d}(p, c_2)$ holds on the ICON grid, the above-mentioned mass flux consistency is ensured. It is noted that attempts to use higher-order polynomial interpolation methods, which are the standard in mesoscale models with regular quadrilateral grids, were unsuccessful on the triangular ICON grid, because the ensuing equation system leads to the inversion of nearly singular matrices.

Rather than interpolating $v_n$ and its time tendency, only the time tendency is interpolated, and then used to update $v_n$ at child level at every dynamics time step. The wind field $v_n$ itself is interpolated only once during the initialization of the child domain. This methodology has been chosen because the comparatively inaccurate interpolation to the interior child edges tends to induce small-scale noise in $v_n$. To suppress the remaining noise arising from the interpolation of the time tendency, a second-order diffusion operator is applied in the inner half of the boundary interpolation zone on $v_n$, and the default fourth-order diffusion applied in the prognostic part of the model domain (Zängl et al., 2015) is enhanced in up to five grid rows adjacent to the interpolation zone by an amount exponentially decaying with distance from the boundary. For the second-order diffusion, a coefficient of $0.005\,a_e/\Delta t$ is applied, where $a_e$ is the area represented by the current triangle edge, and the scaling with $\Delta t$ means that the amount of diffusion is effectively independent from the time step. This proved to be sufficient to suppress the development of spurious disturbances even in the data assimilation cycle, which turned out to be the most critical application mode in this respect. For the other prognostic variables, no special filtering is applied near nest boundaries. In the case of one-way nesting, the second-order velocity diffusion is extended into the nudging zone of the nested domain, replacing the enhanced fourth-order diffusion. More details on the nudging zone are given in Sect. 2.2.3.

For the horizontal mass flux $\rho v_n$, the time average over the dynamic sub-steps is interpolated instead of time level $n$, as the time averaged mass flux is used by the tracer transport scheme in order to achieve consistency with continuity. Using the mass flux time tendency that is interpolated as well, the related time shift is corrected for when applying the boundary mass fluxes at the child level. In the nested domain, the interpolated mass fluxes valid for the current time step are then prescribed at the interface edges separating the boundary interpolation zone from the prognostic part of the nested domain (edges nr. 9 in Fig. 1). Due to the flux-form scheme used for solving the continuity equation (Zängl et al., 2015), this implies that the interpolated

values of $\rho$ do not enter into any prognostic computations in the dynamical core. They are needed, however, for flux limiter computations in the transport scheme. Moreover, no mass fluxes at interior child edges are used, so that the non-conservative interpolation method used for those edges does not affect the model's conservation properties. For $\theta_v$ and the tracer variables $q_k$, the values at the edges are reconstructed in the usual manner following Eq. (20) in Zängl et al. (2015) and then multiplied

with the interpolated mass fluxes before computing the flux divergences.

### 2.2.2   Feedback: Child → Parent

If two-way nesting is activated, the model state $\mathcal{M}_p^{n+1}$ on the parent domain is relaxed towards the updated model state $\mathcal{M}_c^{n+1}$ on the child domain at every parent fast physics time step $\Delta t_p$. This relaxation-type feedback is only applied to the prognostic variables $v_n$, $w$, $\theta_v$, $\rho$ plus specific humidity $q_v$ and the specific contents of cloud water $q_c$ and cloud ice $q_i$. Precipitating

hydrometeors are excluded because recommended relaxation time scales (see below) are larger than their typical falling times. Surface variables are excluded as well because they can easily adjust during runtime, and the tile approach used in ICON's land-surface module would require a rather complicated algorithm to avoid inconsistencies.

Let $\psi$ denote any of the above variables. Conceptually, the feedback mechanism is based on the following three steps:

1. *Upscaling*: The updated variable $\psi_c^{n+1}$ is interpolated (upscaled) from the child domain to the parent domain. The

upscaling operators for cell-based and edge-based variables will be denoted by $\mathcal{I}_{c\to p}$ and $\mathcal{I}_{c\to p}^e$, respectively.

2. *Difference computation*: The difference between the parent-domain variable $\psi_p^{n+1}$ and the upscaled child variable $\mathcal{I}_{c\to p}(\psi_c^{n+1})$ is computed.

3. *Relaxation*: The variable on the parent domain is relaxed towards the upscaled child-domain variable by an increment that is proportional to the difference computed in step 2.

For edge-based normal velocity $v_n$, the arithmetic average of the two child edges lying on the parent edge is taken.

$$\mathcal{I}_{c\to p}^e(v_{n,c}) = \frac{1}{2}\left[v_{n,e_{\text{child 1}}} + v_{n,e_{\text{child 2}}}\right] \tag{3}$$

For cell-based variables the upscaling consists of a modified barycentric interpolation from the four child cells to the corresponding parent cell:

$$\mathcal{I}_{c\to p}(\psi_c) = \sum_{j=1}^{4} \alpha_j \psi_{c_j}. \tag{4}$$

The weights $\alpha_j$ are derived from the following constraints (5)–(7). First of all, a necessary property for the interpolation operator is that it reproduces constant fields, i.e. the weights are normalized:

$$\sum_{j=1}^{4} \alpha_j = 1. \tag{5}$$

Moreover, the interpolation shall be linear: With the four child cell circumcenters $\boldsymbol{x}_j$ $(j = 1, \ldots, 4)$, and $\boldsymbol{x}_p$ denoting the parent cell center, i.e. the interpolation target, we set

$$\sum_{j=1}^{4} \alpha_j(\boldsymbol{x}_j - \boldsymbol{x}_p) = 0. \tag{6}$$

To motivate this constraint, consider the special case of equilateral triangles in which the center point of the inner child cell $x_1$ coincides with the parent center such that the term $(x_1 - x_p)$ vanishes. Equation (6) then defines a barycentric interpolation within the triangle spanned by the mass points of the three outer child cells $\{c_2, c_3, c_4\}$ (see Fig. 2a), where the weights $\{\alpha_2, \alpha_3, \alpha_4\}$ represent the barycentric coordinates. Generally, the constraints (5) and (6) ensure reversibility in the sense that $\mathcal{I}_{c \to p}$ ($\mathcal{I}_{p \to c}$) returns the original parent cell value irrespective of the reconstructed gradient.

Of course, the contribution of the point $x_1$ closest to the interpolation target is of particular importance. Therefore, the underdetermined system of equations (5), (6) is closed with a final constraint which reads as

$$\alpha_1 = \frac{a_{c_1}}{a_p}, \tag{7}$$

where $a_{c_1}$ and $a_p$ denote the inner child and parent cell areas, respectively. In other words, the inner child cell $c_1$ containing the parent cell circumcenter is given a pre-defined weight corresponding to its fractional area coverage. This can be interpreted as a conservation constraint for the special case of a very localized signal at the mass point of the inner child cell.

In summary, this method can be regarded as a *modified* barycentric interpolation for the mass points $\{x_2, x_3, x_4\}$, which accounts for $x_1$ as an additional fourth source point. A more stringent barycentric interpolation would require an additional triangulation based on the child mass points.

We note that the cell-based operator $\mathcal{I}_{c \to p}$ is not strictly mass-conserving and that strict mass conservation would require some means of area-weighted aggregation from the child cells to the parent cells, which is available as an option. The difficulty with such methods on the ICON grid is related to the fact that the mass points lie in the circumcenter rather than the barycenter of the triangular cells. These points coincide on planar equilateral triangles, but they do not on general spherical triangles, the differences being largest in the vicinity of the pentagon points. Due to this fact, using an area-weighted aggregation from the child cells to the parent cells would map linear horizontal gradients on the child grid into a checkerboard noise pattern between upward and downward oriented triangles on the parent grid.

Another difficulty that was encountered in the context of mass conservation is related to the fact that the density decreases roughly exponentially with height. In the presence of orography, the atmospheric mass resolved on the model grid therefore increases with decreasing mesh size, assuming the usual area-weighted aggregation of the orographic raw data to the model grid. Feeding back $\rho$ is thus intrinsically non-conservative. To keep the related errors small and non-systematic, and to generally reduce the numerical errors over steep mountains, perturbations from the reference state are used for upscaling $\rho$ and $\theta_v$ to the parent grid. A closer investigation of the related conservation errors revealed that the previously mentioned differences between bilinear and area-weighted averaging are negligible (with real orography) compared to the mesh-size-related conservation error.

When combining the above-mentioned steps, the feedback mechanism for $\rho$ can be cast into the following form:

$$\rho_p^* = \rho_p^{n+1} + \frac{\Delta t_p}{\tau_{fb}} \left( \mathcal{I}_{c \to p}(\rho_c^{n+1} - \Delta \rho_{\text{corr}}) - \rho_p^{n+1} \right). \tag{8}$$

Here $\rho_p^{n+1}$ denotes the parent-cell density, which has already been updated by dynamics and physics. The superscript "$*$" signifies the final solution including the feedback increment. $\Delta t_p$ is the fast physics time step on the parent domain, and $\tau_{fb}$ is a user-defined relaxation time scale that has a default value of $\tau_{fb} = 10800\,\text{s}$ and is independent of the relaxed field. The

smaller $\tau_{fb}$, the faster the parent state is drawn towards the child state. The chosen default value is optimized for our typical NWP applications and aims at filtering small-scale transient features from the feedback, while fully capturing synoptic-scale features.

Finally note that the upscaled density includes the correction term $\Delta\rho_{\mathrm{corr}}$, which has been introduced to account for height differences between the child and parent cell circumcenters. At locations with mountainous orography, the heights at parent cell circumcenters can differ markedly from those at the corresponding child cells. Without an appropriate correction, the feedback process would introduce a noticeable bias in the parent domain's mass field. The correction term is given by

$$\Delta\rho_{\mathrm{corr}} = \left(1.05 - 0.005\,\mathcal{I}_{c\rightarrow p}(\theta'^{n+1}_{v,c})\right)\Delta\rho_{\mathrm{ref},p}\,,$$

with the parent-child difference of the reference density field

$$\Delta\rho_{\mathrm{ref},p} = \mathcal{I}_{c\rightarrow p}(\rho_{\mathrm{ref},c}) - \rho_{\mathrm{ref},p}\,,$$

and the potential temperature perturbation $\theta'^{n+1}_{v,c} = \theta^{n+1}_{v,c} - \theta_{v\,\mathrm{ref},c}$. The term $\Delta\rho_{\mathrm{ref},p}$ is a pure function of the parent-child height difference and can be viewed as a leading-order correction term. As a further optimization, the empirical factor $(1.05 - 0.005\,\mathcal{I}_{c\rightarrow p}(\theta'^{n+1}_{v}))$ was added, which means that the correction is roughly proportional to the actual air density. We further note that a possibly more accurate and less ad hoc approach would require a conservative remapping step in the vertical, prior to the horizontal upscaling.

Care is required in order to achieve consistency with continuity. For this purpose, feedback is not implemented for the tracer mass fractions directly, but for partial densities. Building upon the implementation for $\rho$, we get

$$(\rho q_k)^*_p = (\rho q_k)^{n+1}_p + \frac{\Delta t_p}{\tau_{fb}}\left[\mathcal{I}_{c\rightarrow p}((\rho^{n+1}_c - \Delta\rho_{\mathrm{corr}})q^{n+1}_{k,c}) - (\rho q_k)^{n+1}_p\right] \tag{9}$$

Mass fractions are re-diagnosed thereafter:

$$q_{k,p} = \frac{(\rho q_k)^*_p}{\rho^*_p}$$

By summing Eq. (9) over all partial densities, we recover Eq. (8) for the total density.

A very similar approach is used for $\theta_v$. As for $\rho$, only the increment of $\theta_v$ is upscaled from the child domain to the parent domain and added to the parent reference profile $\theta_{v\,\mathrm{ref},p}$.

$$\theta^*_{v,p} = \theta^{n+1}_{v,p} + \frac{\Delta t_p}{\tau_{fb}}\left(\mathcal{I}_{c\rightarrow p}(\theta'^{n+1}_{v,c}) + \theta_{v\,\mathrm{ref},p} - \theta^{n+1}_{v,p}\right)$$

The same approach is taken for $w$, however the full field is upscaled.

$$w^*_p = w^{n+1}_p + \frac{\Delta t_p}{\tau_{fb}}\left(\mathcal{I}_{c\rightarrow p}(w^{n+1}_c) - w^{n+1}_p\right)$$

In the case of $v_n$ some second-order diffusion is added to the ensuing feedback increment in order to damp small-scale noise.

$$v^*_{n,p} = v^{n+1}_{n,p} + \frac{\Delta t_p}{\tau_{fb}}\left(\Delta v_{n,p} + K\,\nabla^2\left(\Delta v_{n,p}\right)\right)\,, \tag{10}$$

with the feedback increment

$$\Delta v_{n,p} = \mathcal{I}^e_{c \to p}(v^{n+1}_{n,c}) - v^{n+1}_{n,p},$$

and the diffusion coefficient $K = \frac{1}{12} \frac{a_{p,e}}{\Delta t_p}$, where $a_{p,e}$ is the area of the quadrilateral spanned by the vertices and cell centers adjacent to the parent's edge.

### 2.2.3 Lateral Nudging

If the feedback is turned off, i.e. if one-way nesting is chosen, a nudging of the prognostic child-grid variables towards the corresponding parent-grid values is needed near the lateral nest boundaries in order to accommodate possible inconsistencies between the two grids, particularly near the outflow boundary. Nudging is performed every fast physics time step of the child domain $\Delta t_c$ following Davies (1976), by adding a forcing term to the prognostic equations for $v_n$, $\rho$, $\theta_v$, and $q_v$ of the form

$$\frac{\partial \psi_c}{\partial t} = RHS + \frac{1}{\Delta t_c} \alpha_{\text{nudge}} \left[ \mathcal{I}_{p \to c} \left( \psi_p - \mathcal{I}_{c \to p}(\psi_c) \right) \right],$$

with the nudging coefficient $\alpha_{\text{nudge}}$, and the term in brackets denoting the nudging increment. Because lateral boundaries are in general not straight lines on the unstructured ICON grid, attempts to make an explicit distinction between inflow and outflow boundaries (e.g. by prescribing $v_n$ at inflow boundaries only) were not successful.

To compute the nudging increment, the child-grid variables are first upscaled to the parent grid in the same way as for the feedback (Eq. (3) and (4)), followed by taking the differences between the parent-grid variables and the upscaled child-grid variables. The differences are then interpolated back to the child grid using the same methods as for the lateral boundary conditions (Eq. (1) and (2)). The nudging coefficient $\alpha_{\text{nudge}}$ decreases exponentially from the inner margin of the boundary interpolation zone towards the interior of the model domain and is defined as

$$\alpha_{\text{nudge}} = \begin{cases} A_0 \exp\left(-\frac{r-r_0}{\mu}\right), & \text{if } r > r_0 \text{ and } r - r_0 \leq L \\ 0 & \text{otherwise}, \end{cases}$$

with the maximum nudging coefficient $A_0$, the cell row index $r$, the nudging zone start index $r_0 = 5$ (see Fig. 1), the nudging zone width $L$ and the e-folding width $\mu$, the latter two defined in units of cell rows. The coefficients $A_0$, $L$, and $\mu$ may be adjusted by the user, with the default values given by $A_0 = 0.1$, $L = 8$ cell rows and $\mu = 2$ cell rows. A second-order diffusion on $v_n$ is used near the lateral nest boundaries in order to suppress small-scale noise. As opposed to Eq. (10) for feedback, the diffusion operator acts on the velocity field itself, rather than the velocity increment.

## 2.3 Vertical Nesting

The vertical nesting option allows to set model top heights individually for each domain, with the constraints that the child domain height is lower than or at most equal to the parent domain height, and that the child domain extends into heights where the coordinate surfaces are flat. This allows, for instance, a global domain extending into the mesosphere to be combined with

a child domain that extends only up to the lower stratosphere, which can save a significant amount of computational resources. However, a vertical refinement in the sense that the vertical resolution in the child domain may differ from that in the parent domain is not implemented.

Vertical nesting requires appropriate boundary conditions for all prognostic variables to be specified at the vertical nest interface level, i.e. the uppermost half level of the nested domain. This is crucial in order to prevent vertically propagating

sound and gravity waves from being spuriously reflected at the nest interface. In the following, boundary conditions are derived for $v_n$, $w$, $\theta_v$, $\rho$, $q_k$ as well as the vertical mass flux $\rho w$. We note that the boundary condition for $w$ is required for the vertically implicit sound wave solver in the dynamics, whereas $\rho w$ is needed to compute the vertical flux divergence terms in the prognostic equations for $\rho$, $\pi$ and $\rho q_k$.

Due to the constraints mentioned above, boundary conditions can be derived by horizontal parent-to-child interpolation,

without the need of any boundary interpolation zone extending vertically away from the upper nest boundary. For $w$, $\theta_v$, $\rho$ and $\rho w$ the full fields at the nest interface level are horizontally interpolated from the parent to the child grid, using the same RBF based interpolation method as for the lateral boundary conditions (Eq. (1)). Rather than interpolating instantaneous values as for the lateral boundaries, $w$, $\theta_v$, $\rho$, and $\rho w$ are averaged over all dynamics substeps constituting a fast physics time step, in order to filter oscillations related to vertically propagating sound waves. Hence, for $\psi \in \{w, \theta_v, \rho, \rho w\}$ the parent-to-child

interpolated field reads

$$\overline{\psi}_c = \mathcal{I}_{p \to c} \left( \frac{1}{\texttt{nsubs}} \sum_{s=1}^{\texttt{nsubs}} \psi_p^{n+s/\texttt{nsubs}} \right), \tag{11}$$

with $s$ denoting an individual dynamics substep, and $\texttt{nsubs}$ denoting the total number of substeps. We apply the same interpolation method to the corresponding time tendencies $\partial \psi_p / \partial t$, which are estimated by taking the difference of the state variables at the substeps $s=1$ and $s=\texttt{nsubs}$. This enables us to perform a linear interpolation in time, in order to provide the boundary

conditions at approximately the right time levels for every dynamics substep on the child domain.

A slightly different approach is taken for $v_n$, which turned out to be beneficial in order to reduce the magnitude of the horizontal interpolation errors. The differences between the nest interface level and the next half level below (denoted as $\Delta v_{n,p}$ in the following) are interpolated rather than the full field, using again the same methods as for the lateral boundary conditions. After interpolating $\Delta v_{n,p}$ to the child domain, it is added to $v_{n,c}$ at the second interface level ($k=3/2$) on the child domain,

in order to obtain the upper boundary condition, i.e.

$$v_{n,c}(k=1/2) = v_{n,c}(k=3/2) + \mathcal{I}_{p \to c} \left( \frac{1}{2} \left( \Delta v_{n,p}^n + \Delta v_{n,p}^{n+1} \right) \right).$$

Since $\Delta v_n$ is less strongly affected by sound waves, only an average between the first and the last dynamics substep is taken prior to the interpolation. The temporal interpolation is neglected.

For the tracer variables we refrain from interpolating the partial mass fluxes $(\rho w q_k)_p$ directly, in order to ensure tracer- and

air mass consistency. Instead, we make use of the vertical mass flux boundary condition $(\overline{\rho w})_c$ and multiply it with proper mass fractions. On the parent domain the required mass fractions are derived by taking the ratio of the vertical tracer mass flux at the nest interface level calculated in the vertical tracer transport scheme $(\rho w q_k)_p$ and the available mass flux $(\rho w)_p$. The mass

fractions are then interpolated to the child domain, using Eq. (1). Hence, the flux boundary condition for an arbitrary tracer field $q_k$ reads


$$(\rho w q_k)_c = (\overline{\rho w})_c \, \mathcal{I}_{p \to c} \left( \frac{(\rho w q_k)_p}{(\rho w)_p} \right) ,$$

with $(\overline{\rho w})_c$ computed via Eq. (11). We note that due to the lack of $q_k$ values above the nest upper boundary (lack of a boundary interpolation zone), the flux computation for scalars at the second interface level ($k = 3/2$) is only stable for vertical Courant numbers $C_k = |w_k| \Delta t_c / \Delta z_k|_{k=3/2} \leq 1$, with $\Delta z$ denoting the vertical layer thickness.

## 2.4 Recursive Algorithm for Multi-domain Setups

The previous sections have focused on the coupling of a single child domain. The nesting capability of ICON, however, is not limited to a single domain but supports multiple nests at the same level and multi-level nesting, as well as a combination of both. In the literature, multi-level nesting is also referred to as telescoping nesting (Mouallem et al., 2022). An example of multiple same-level nesting will be provided in Sect. 3.1, while for multi-level nesting we refer to the ICON simulations in Weimer et al. (2021), where a three-domain three-level setup has been used to investigate mountain-wave induced polar

stratospheric clouds.

The coupling of multiple same-level nested domains with a parent domain is rather straightforward, as it only requires the single-nest coupling strategy (Sect. 2.2.1–2.2.2) to be applied sequentially for each nest. The coupling strategy for repeatedly nested domains is probably less obvious and will be described here for clarity, and in order to complement already existing applications of this feature (e.g. Weimer et al., 2021).

Figure 3 displays a basic multi-level nesting example, where a global domain is combined with two successively (two-way) nested domains. The global domain is indicated at the bottom, and the nested domains are vertically staggered on top of it. The red and blue regions show the boundary interpolation zones and feedback zones of the individual domains, respectively. The integration time step on the global domain is denoted by $\Delta t$. It is automatically reduced by a factor of 2 when progressing to the next child grid level.

The whole processing sequence for integrating all domains from time step $n$ to $n + 1$ is shown in the flowchart at the lower left of Fig. 3. The domains are ordered top down. Open and filled black dots show model states without and with feedback increments included, black arrows indicate time integration with the fast physics time step, and red and blue arrows indicate lateral boundary data interpolation and feedback, respectively.

The flow control of ICON's hierarchical nesting scheme is handled by a recursive subroutine that cascades from the global

domain (or outer limited-area domain) down to the deepest nesting level and calls the dynamical core and the physics parameterizations for each domain in basically the same way as for the global domain. The basic processing sequence is as follows:

1. A single integration step with $\Delta t$ is executed on the global domain, resulting in an updated model state $\mathcal{M}_p^{n+1}$, as indicated by an open black circle in Fig. 3.

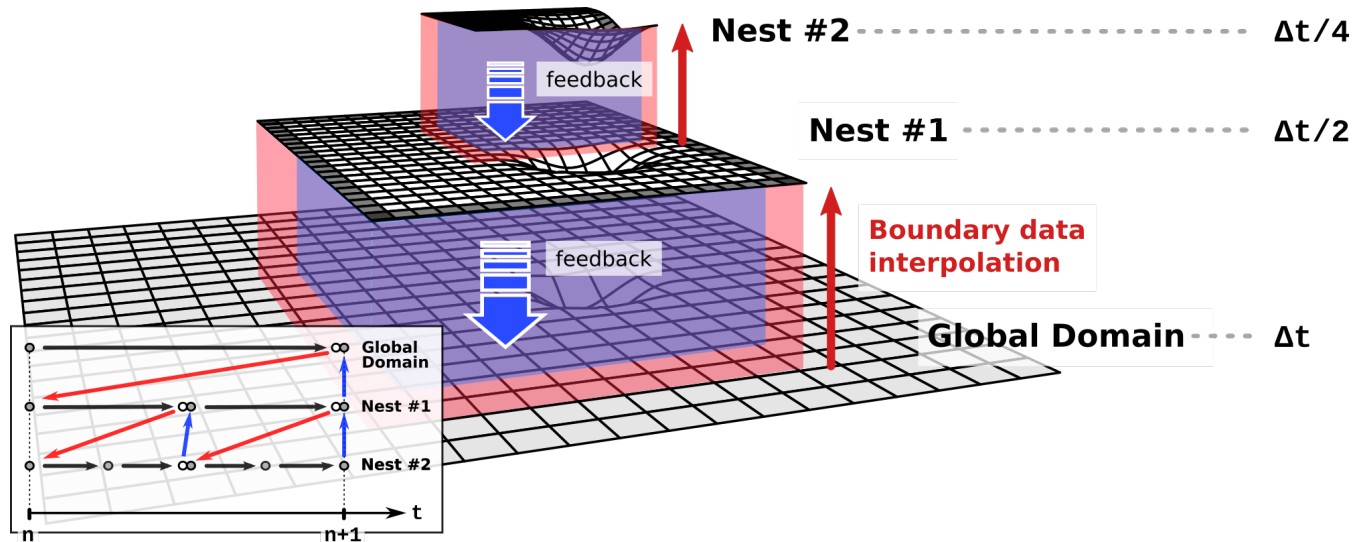

**Figure 3.** Schematic of a multi-domain multi-level setup with two domains nested successively into a global (or limited-area) base domain. The processing sequence for the time integration of all domains from time step $n$ to $n+1$ is shown in the flowchart at the lower left. See the text for details.

2. Boundary data are interpolated from the global domain to nest 1 (red arrow), followed by an integration step on nest 1 over the time interval $\Delta t/2$, and resulting in the model state $\mathcal{M}_{c1}^{n+1/2}$.

3. As there exists another nested domain within nest 1, boundary fields based on the model state $\mathcal{M}_{c1}^{n+1/2}$ are interpolated to the second nested domain. Afterwards, the model is integrated on nest 2 over two times the time interval $\Delta t/4$, resulting in the model state $\mathcal{M}_{c2}^{n+1/2}$.

4. Feedback is conducted from nest 2 back to nest 1 (blue arrow), resulting in an updated model state $\mathcal{M}_{c1}^{n+1/2*}$ on nested domain 1 (black filled dot). Then, on the nested domain 1 the model is again integrated in time to reach model state $\mathcal{M}_{c1}^{n+1}$.

5. This is followed by a second lateral boundary data interpolation from nest 1 to nest 2 based on $\mathcal{M}_{c1}^{n+1}$. Nest 2 is integrated in time again, to reach its state $\mathcal{M}_{c2}^{n+1}$.

6. As a final step, feedback is performed from nest 2 to nest 1, followed by feedback from nest 1 to the global domain.

We note that the presented coupling strategy is very similar to that in WRF or FV3, but differences exist in several details. For example, the child-to-parent feedback in FV3 covers only temperature and the wind components (Mouallem et al., 2022), which avoids any impact on mass conservation but apparently necessitates executing the feedback before the physics call at the parent level in order to maintain numerical stability.

 **3  Application Examples**

### 3.1  Jablonowski-Williamson Test

To demonstrate the functionality of the grid nesting and to investigate the numerical errors related to the mesh size discontinuity along the nest boundary, we start with the Jablonowski and Williamson (2006) test (named JW test hereafter) already used in Zängl et al. (2015) and many other studies to evaluate basic aspects of numerical accuracy. This test considers the formation of baroclinic waves in a geostrophically and hydrostatically balanced zonally symmetric basic state with a very strong equator-to-pole temperature contrast. In its standard version, the waves are triggered by a small perturbation in the initial wind field imposed at 20°N, 40°E. The perturbation grows very slowly during the first few days but evolves into an explosive cyclogenesis between days 7 and 10 of the test case. Dropping the initial perturbation, which implies that the model ideally should maintain the initial state, allows for testing the numerical errors related, for instance, to grid irregularities.

**Table 1.** List of model configurations for the Jablonowski-Williamson test.

| Exp. ID | global grid | position of 1st nest | position of 2nd nest |
|---------|-------------|----------------------|----------------------|
| E1 | R2B4 | — | — |
| E2 | R2B4 | 32.5°N–72.5°N, 65°E–115°E | — |
| E3 | R2B4 | 32.5°N–72.5°N, 65°E–115°E | 32.5°N–72.5°N, 175°W–125°W |
| E4 | R2B4 | 32.5°N–72.5°N, 30°E–130°W | — |
| E5 | R2B5 | — | — |

Our first series of experiments considers the baroclinic wave test for a variety of configurations based on the model grids R2B4 (mesh size $160\,\text{km}$) and R2B5 ($80\,\text{km}$). Experiment E1 uses the global R2B4 grid only, experiments E2–E4 use a global R2B4 grid plus nested R2B5 grids at various locations along the track of the baroclinic wave (see Table 1 for details), and E5 uses a global R2B5 grid for reference. To account for the fact that these resolutions are much coarser than in typical NWP applications, the relaxation time scale $\tau_{fb}$ is increased to 12 h. The results are summarized in Fig. 4, showing the surface pressure and the relative vorticity at $850\,\text{hPa}$. For the nested experiments E2–E4, the results are displayed for the coarse (R2B4) grid, implying that the impact of the nested domain(s) appears indirectly via the feedback mechanism described above. Selected results for the nested domain are added in Fig. 5, while combined results are displayed in Fig. 6, where the highest resolution data is used in any region of the plot.

As discussed in Zängl et al. (2015), the general behaviour of ICON (and most other grid-point models) in the JW test is that the baroclinic wave train exhibits a phase lag at coarse resolution. Moreover, the leading cyclone tends to be too weak at coarse resolution, whereas the trailing cyclone tends to be slightly too intense because grid irregularities act as an additional trigger for wave-like disturbances. In Fig. 4, these features become evident from comparing E1 (Fig. 4a) with E5 (Fig. 4i). Comparing the nested runs E2 and E3 (Fig. 4c,e) with E1 reveals that the western nested domain, which is shared by E2 and E3 and is crossed by the baroclinic wave during the early stage of its evolution (days 2–4), leads to a slight reduction of the phase lag

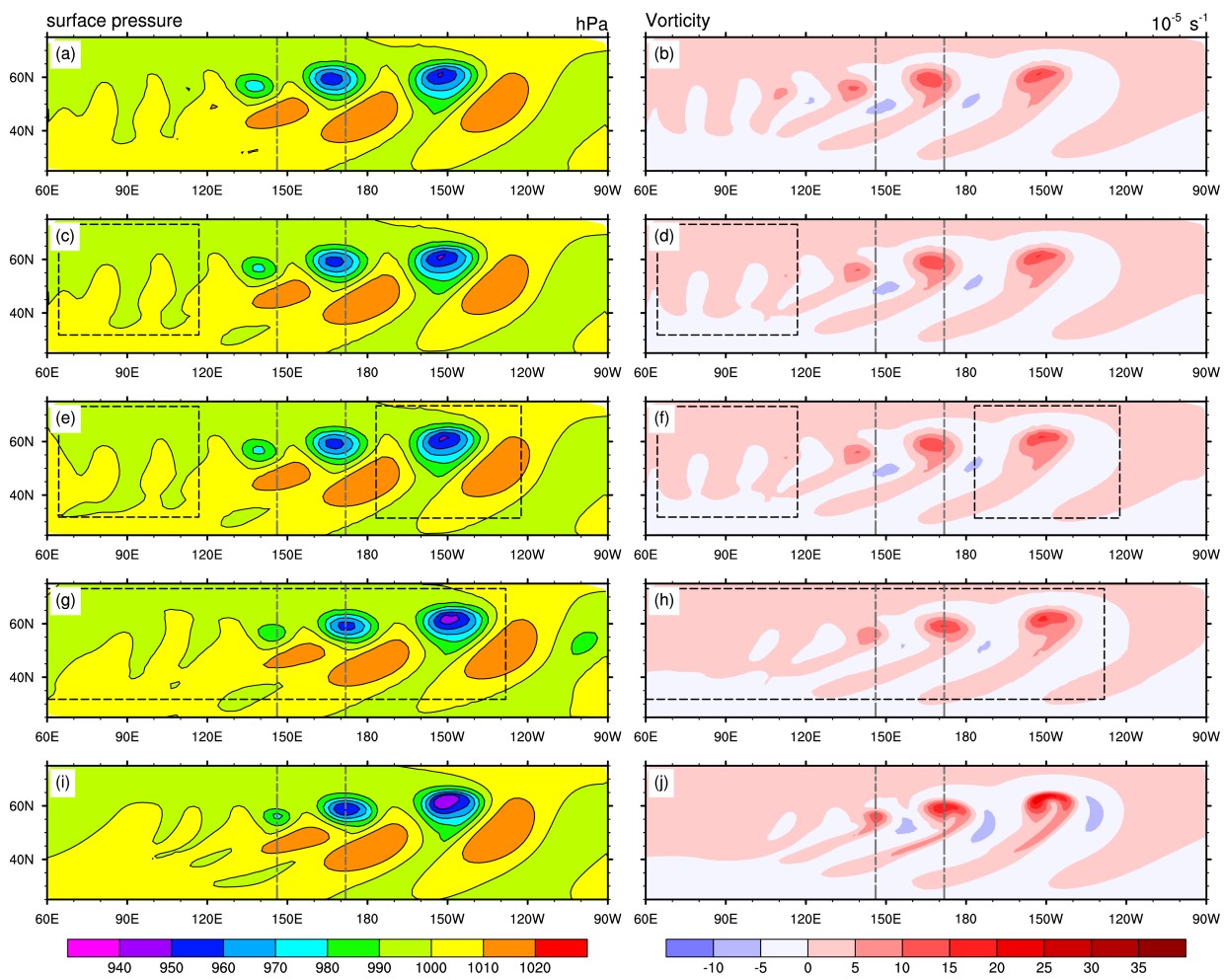

**Figure 4.** Surface pressure (hPa, left column) and relative vorticity at $850\,\text{hPa}$ ($10^{-5}\text{s}^{-1}$, right column) for the Jablonowski-Williamson experiments E1 (a,b), E2 (c,d), E3 (e,f), E4 (g,h), and E5 (i,j) after 9 days of integration. Dashed boxes show the nest locations. To ease comparison, the dashed gray line indicates the longitudinal position of the trailing cyclone's minimum surface pressure in the R2B5 reference run (E5). See Table 1 for the experiment configurations.

and the intensity bias of the trailing cyclone. On the other hand, there is very little difference between E2 and E3, implying that the eastern nested domain entered by the leading cyclone at the beginning of its rapid intensification around day 7 has only a weak impact on the cyclone intensity. More pronounced differences appear with the large nested domain of E4, covering the baroclinic wave train during most of its life cycle. In the nested domain (see Fig. 6), the position and intensity of the cyclones is now very similar to E5. The relaxation-based feedback to the coarse domain shown in Fig. 4g shifts the cyclones to the right positions, but their intensity is weaker than in the reference experiment E5. The intensity difference is primarily related to the

above-mentioned long relaxation time scale of 12 hours, combined with the fact that the middle and left cyclones propagate at

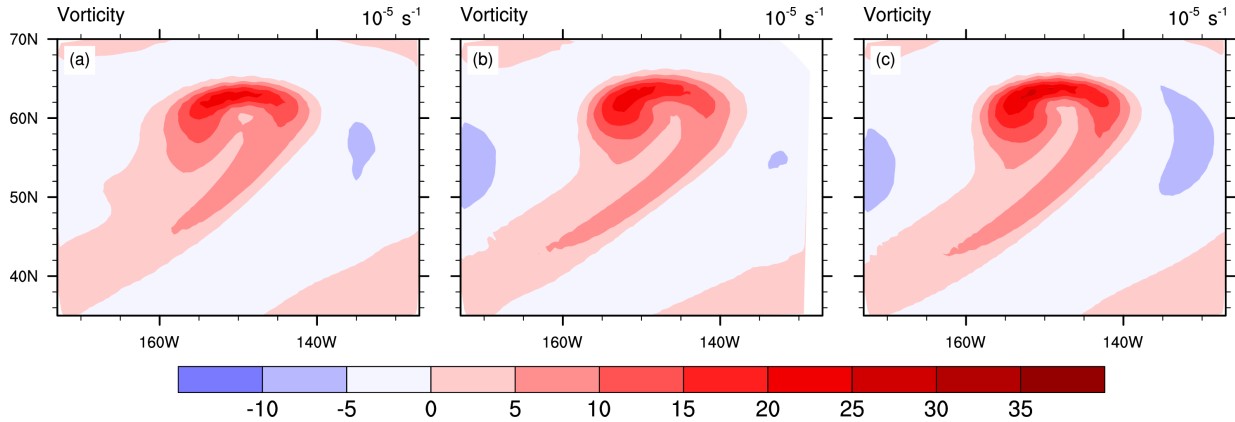

**Figure 5.** Relative vorticity at $850\,\mathrm{hPa}$ ($10^{-5}\mathrm{s}^{-1}$) after 9 days of integration for experiments E3 (a), E4 (b), and E5 (c), calculated at R2B5 resolution in each case (i.e. in the nested domain for E3 and E4).

a somewhat slower speed in the coarse domain. Reducing $\tau_{fb}$ gradually decreases the difference, but for values less than about $5\Delta t_p$ (i.e. two hours), the solution in the nested domain starts to deviate from the E5 reference because numerical disturbances generated along the nest boundaries become non-negligible (not shown). Note in this context that in typical NWP applications, the resolution in the global domain is always high enough that cyclones and fronts move at the same speed as in the nested domain(s) and possess about the same intensity (except for tropical cyclones), so that the intensity bias encountered here is not of practical relevance. On the other hand, the standard relaxation time scale of three hours filters transient wave-like motions that are not deterministically predictable anyway, thus avoiding a transfer of small-scale noise into the global domain.

The relative vorticity fields (right panels of Fig. 4) confirm that the nested domain in E2 (and E3) reduces the phase lag of the trailing cyclone as well as the intensity of an extra disturbance on the upwind side that does not appear in E5. However, the intensity differences between E5 and the other experiments appear much more pronounced than in the surface pressure field. This is primarily because the accuracy at which derivative-based quantities like vorticity can be calculated depends directly on the mesh size of the model grid. To obtain better comparability, Fig. 5 displays vorticity fields for the leading cyclone computed at the same mesh size (R2B5) for E3, E4 and E5. Compared to the reference result from E5 (Fig. 5c), the E3 vorticity field exhibits significant distortions and a somewhat reduced amplitude (Fig. 5a), reflecting the fact that the cyclone was only poorly resolved during part of its development stage. For E4 (Fig. 5b), the differences are much smaller and concentrate to the vicinity of the nest boundary at 130°W. Note that there are no evident disturbances near the transition between the lateral boundary interpolation zone and the prognostic computational domain of the nested domain at about 135°W. Further, there is no evidence of significant discontinuities along the parent-child domain interface, as can deduced from the combined solution plot in Fig. 6.

To further examine the flow disturbances generated by the resolution jump at the nest boundary, Fig. 7 compares the surface pressure fields for the steady-state JW test after 9 days of integration for E1 and E2. This variant of the JW test accentuates the so-called grid imprinting errors arising from any irregularity in the model grid, in our case both due to the non-uniformity of the icosahedral grid and due to the resolution jump along the nest boundaries. As a side note we mention that previous

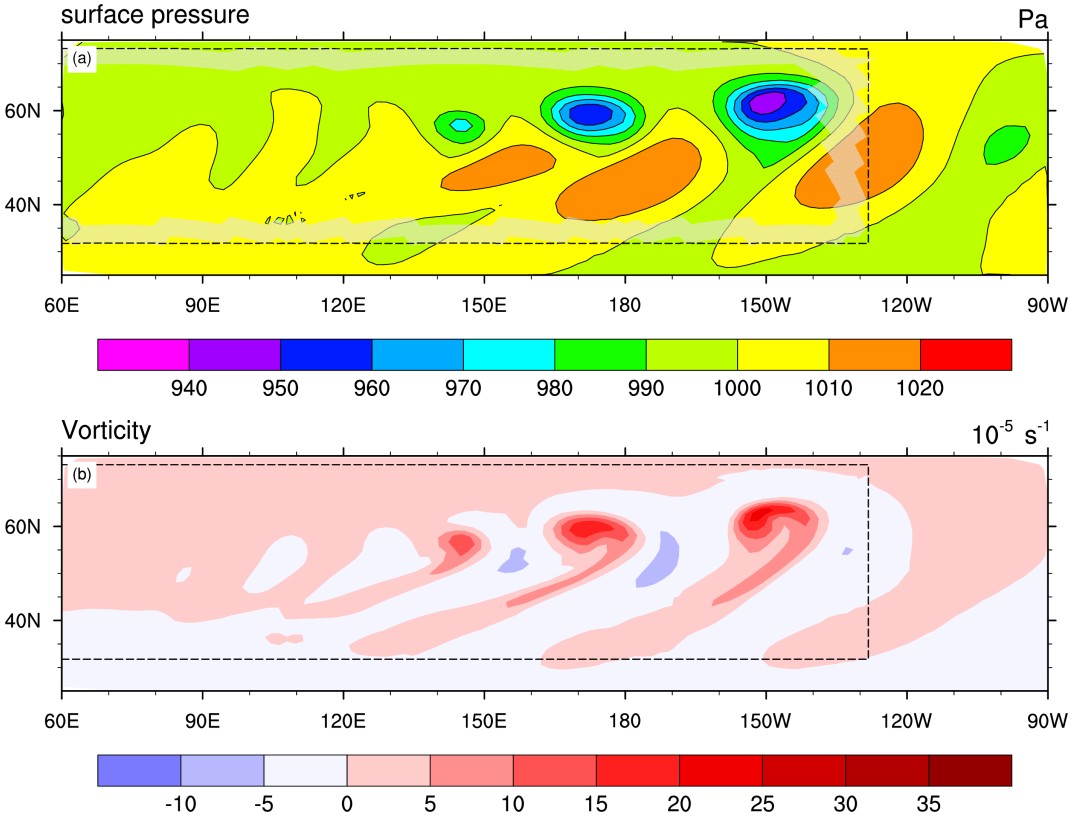

**Figure 6.** (a) Surface pressure and (b) relative vorticity at $850\,\mathrm{hPa}$ ($10^{-5}\mathrm{s}^{-1}$) for JW experiment E4. The combined solution is plotted which means that the highest resolution data is used in any region of the plot. The dashed box shows the location of the nest and the white-shaded region in (a) highlights the exact location of the non-prognostic boundary interpolation zone.

non-nested ICON results presented in Lauritzen et al. (2010) were obtained with an early version of the hydrostatic dynamical
core (Wan et al., 2013), which did not perform as well as the nonhydrostatic dynamical core presented here. The result for
E1 (Fig. 7a) shows the well-known regular wavenumber-five disturbance pattern characteristic for icosahedral grids (see e.g.
Jablonowski and Williamson, 2006; Lauritzen et al., 2010). With the presence of a nested domain, the disturbances become
irregular and reach a somewhat stronger peak amplitude (Fig. 7b). The largest disturbances occur in the air mass that was
initially located in the nest region because the nest-induced perturbations there have the longest time to grow. For smaller
values of $\tau_{fb}$, the amplitude of these disturbances gradually increases (not shown). Regarding their practical relevance, we
note that the baroclinicity of the real atmosphere is generally much weaker than in the initial state of the JW test because it is
continuously depleted by synoptic-scale disturbances. Moreover, the nest-induced perturbations decrease about proportionally
to the grid imprinting in the global grid (see Fig. 2 in Zängl et al., 2015) when refining the model resolution. In DWD's
operational NWP applications, we have never encountered noticeable artificial disturbances along nest boundaries.

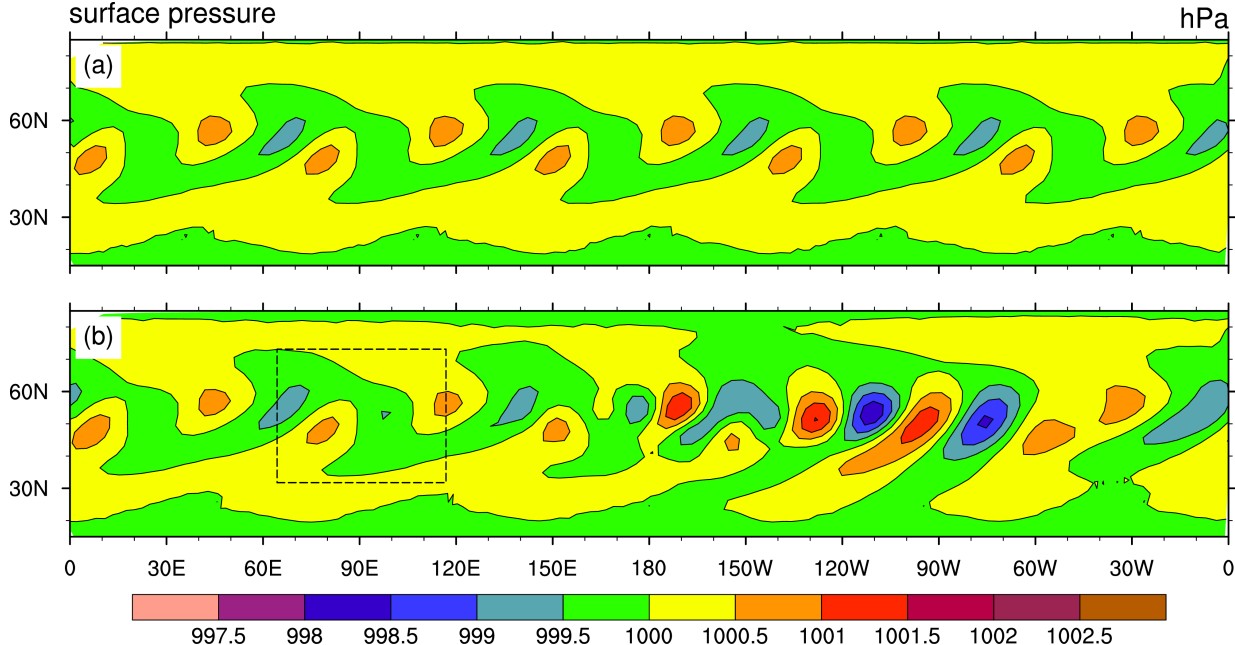

**Figure 7.** Surface pressure (hPa) after 9 days of integration for the steady-state JW test and experiments E1 (a) and E2 (b). The dashed box shows the nest location.

## 3.2  Schär Mountain Test with Vertical Nesting

In order to examine the behaviour of the upper boundary condition for vertically nested domains, we conducted the Schär mountain test case (Schär et al., 2002). Therein, a uniform flow of constant wind speed $U$ and Brunt-Väisälä-frequency $N$ over idealized hilly terrain gives rise to a combination of small-scale, vertically decaying non-hydrostatic gravity waves and larger-scale vertically propagating quasi-hydrostatic gravity waves. We expect this test case to challenge the upper boundary condition implementation, as the vertically propagating waves should pass through the domain interface without spurious reflections or any accumulation of noise.

Following Zängl et al. (2015), we performed a quasi-three-dimensional analogue of this test on a limited area domain, where the original 1D mountain profile is changed to a 2D ridge-like profile that decays towards zero in cross-flow direction, as the lateral domain boundary is approached. In our earlier study, the limited-area setup was chosen in favour of a small planet configuration (Klemp et al., 2015) because it allows for a better control on the upstream flow conditions in the presence of high mountains, and it is retained here for convenience. The ridge-like profile is given by

$$h(x,y) = h_m \exp\left(-\frac{x^2}{a^2}\right) \cos^2 \frac{\pi x}{\lambda} \exp\left(-\frac{\max(0, |y| - \beta)^2}{a^2}\right),$$

with the originally proposed parameter settings $h_m = 250\,\text{m}$, $a = 5000\,\text{m}$, $\lambda = 4000\,\text{m}$, and the ridge length scale $\beta = 10^5\,\text{m}$. The wind speed is set to $U = 10\,\text{ms}^{-1}$, and the Brunt-Väisälä-frequency is given by $N = 0.01\,\text{s}^{-1}$ for $z < 20\,\text{km}$, and gradually

increases to $N = 0.03 \, \mathrm{s}^{-1}$ above. Note that the latter setting differs from the constant $N$ used by Schär et al. (2002) in order to allow for the high model top desired for our test configuration (see below). This does not allow for comparing against an analytic solution, but this is not needed here because model results serving this purpose have already been provided by Zängl et al. (2015).

We have performed two types of simulations, one with and one without a vertically nested domain. The reference simulation without a vertical nest was performed on a single $3° \times 3°$ limited-area domain centered at the equator with a horizontal mesh size of $\Delta x \approx 620 \, \mathrm{m}$ (R2B12), a vertically stretched grid with 50 vertical layers, and a $40 \, \mathrm{km}$ model top. To avoid the reflection of gravity waves at the model top, a Rayleigh damping layer acting on vertical velocity $w$ was applied above $z = 22 \, \mathrm{km}$ (i.e. encompassing the uppermost 10 vertical layers). The simulation was run for $12 \, \mathrm{h}$ until an approximately steady state was reached. The steady-state solution for $w$ is depicted in Fig. 8a. While the wave structure at low levels closely matches previously published results for configurations with constant $N$ (e.g. Schär et al., 2002; Skamarock et al., 2012; Zängl et al., 2015) partial wave reflections probably related to the stability change starting at $z = 20 \, \mathrm{km}$ become apparent in the upper part of the domain.

For the nested simulation, a $2.4° \times 2.4°$ R2B12 domain with 39 vertical layers and a vertical interface at $20 \, \mathrm{km}$ is two-way nested into a $3° \times 3°$ R2B11 parent domain ($\Delta x \approx 1.2 \, \mathrm{km}$) with 50 vertical layers and a $40 \, \mathrm{km}$ model top. There is no Rayleigh damping active in the nested domain. Note that the vertical layer distribution, as well as the horizontal mesh size of the nested domain, exactly match those of the reference simulation. The only significant difference lies in the much lower, but open domain top. Steady-state solutions for the nested domain are depicted in Fig. 8b,c. They differ in terms of the relaxation time scale $\tau_{fb}$ for child-to-parent feedback, with values of $\tau_{fb} = 10800 \, \mathrm{s}$ and $\tau_{fb} = 900 \, \mathrm{s}$ for panels (b) and (c), respectively.

In general, the results for the nested domain and the reference result are fairly close to each other. There is no indication of substantial wave energy reflection or noise accumulation along the nest interface level. Deviations from the reference result are largely confined to the uppermost quasi-hydrostatic wave crest and trough, and to the leeward propagating wave signal. This does not only hold for the steady-state solution, but also for the spinup phase (not shown). The reason for the deviations is twofold: Firstly, the computation of the boundary conditions at the nest interface level inevitably goes along with spatio-temporal interpolation errors. Second, and more importantly, the solutions on the parent and child domains are slightly different, implying that the vertical interface condition derived from the parent domain cannot exactly match the solution on the child domain. These differences primarily originate from the differences in the mesh size, but they additionally depend on the feedback time scale. Reducing the feedback time scale strengthens the domain coupling and reduces the parent-child differences, which in turn improves the vertical nest interface conditions. This can be seen by comparing Fig. 8b ($\tau_{fb} = 10800 \, \mathrm{s}$) with Fig. 8c ($\tau_{fb} = 900 \, \mathrm{s}$), which shows that a shorter feedback time scale further reduces the difference to the non-nested reference (Fig. 8a). The corresponding solution on the R2B11 parent domain (with $\tau_{fb} = 900 \, \mathrm{s}$) is depicted in Fig. 9. Due to its two times coarser mesh size it is lacking some of the leeward propagating wave signals and shows a slightly reduced amplitude of the quasi-hydrostatic wave. The attenuated, though visible, leeward propagating wave signal in Fig. 9 results from the child-to-parent feedback and does not exist in a R2B11 simulation without nest (not shown).

From this test it can be concluded that, even without any interpolation error, the boundary conditions at the nest interface

will never match perfectly due to the resolution-induced differences between the parent and child model states. On the other

hand, this test has shown a small but noticeable positive impact of the child-to-parent feedback mechanism on the quality of

the nest interface conditions, which improve with decreasing feedback timescale.

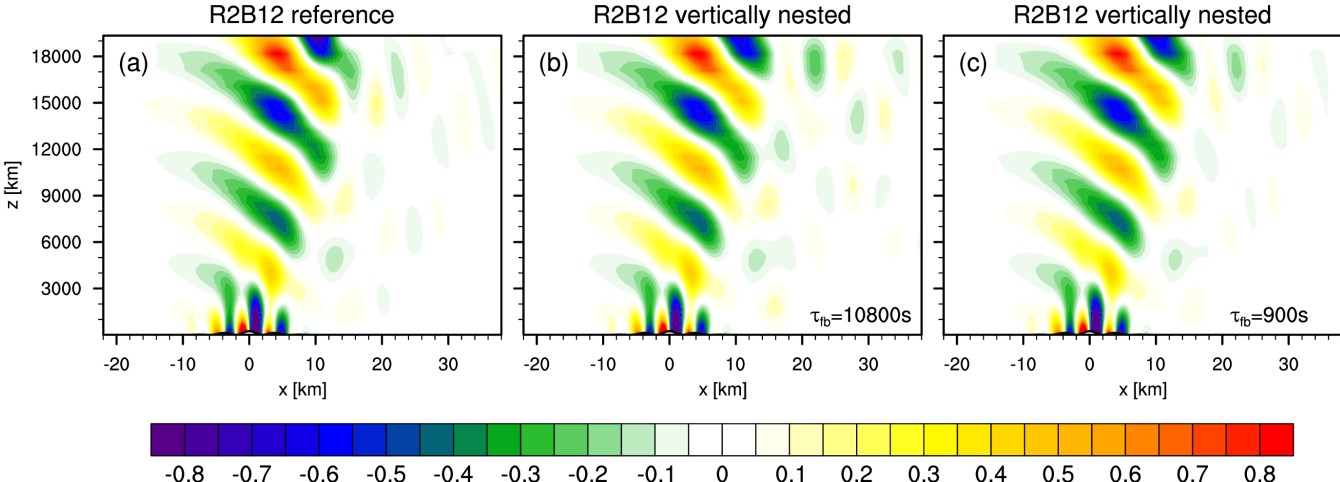

**Figure 8.** Vertical velocity $w$ after $12\,\mathrm{h}$ for the Schär et al. (2002) test case (contour interval $0.05\,\mathrm{ms}^{-1}$). (a) Reference results for a single, non-nested R2B12 domain with $40\,\mathrm{km}$ model top. (b),(c) Results for a vertically nested R2B12 domain with $20\,\mathrm{km}$ model top, and two distinct relaxation time scales for child-to-parent feedback. The entire vertical extent of the nested domain is depicted.

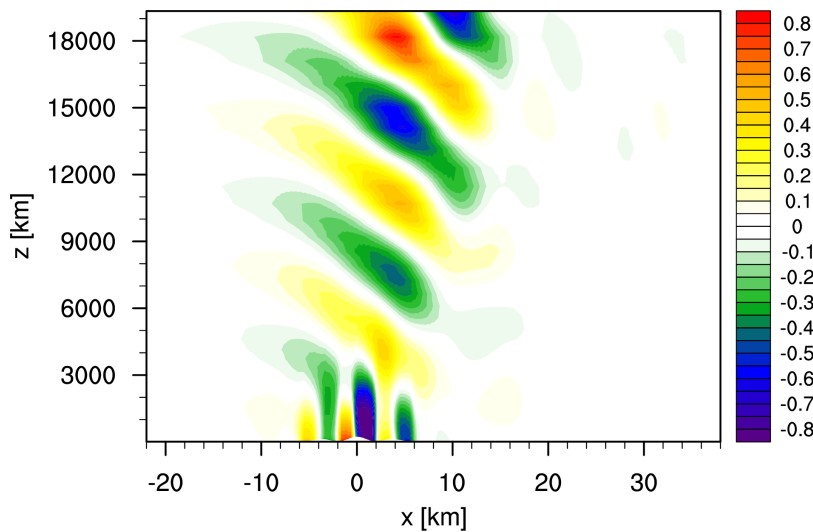

**Figure 9.** Vertical velocity $w$ after $12\,\mathrm{h}$ for the R2B11 parent domain, with $\tau_{fb} = 900\,\mathrm{s}$.

## 3.3 Operational NWP Applications

In an operational context, one ideally expects that the beneficial impact on forecast quality of the regionally refined resolution is transferred to the global domain in the nest overlap area and subsequently propagates downstream, which is usually eastward in the extratropics. This implicitly assumes that the scores used to quantify forecast quality do improve with increasing model resolution, which, according to our experience, is the case for mesh sizes coarser than about $10\,\text{km}$ but not necessarily in the convective gray zone. In the operational global forecasting system of DWD, the deterministic part has a global horizontal mesh size of $13\,\text{km}$ and a two-way nested domain referred to as "ICON-EU" with $6.5\,\text{km}$ mesh size covering Europe and some adjacent regions ($24.5°\text{W} - 63.5°\text{E}$; $29°\text{N} - 71°\text{N}$, i.e. domain 2 in Fig. 1). For this resolution range, the impact of the nested domain on the global forecast quality tends to be rather small. A much clearer impact is found for the corresponding ensemble prediction system (EPS), which (at the time of writing this manuscript) uses mesh sizes of $40\,\text{km}$ and $20\,\text{km}$, respectively. To demonstrate the benefit of two-way nesting on NWP quality, we therefore use the EPS configuration of ICON, but the subsequent experiments are executed as deterministic forecasts for simplicity and easier reproducibility outside the operational environment of DWD. Moreover, our experiments are initialized with interpolated operational IFS analyses available from the European Centre for Medium-Range Weather Forecasts (ECMWF), which has the conceptual advantages that none of our experiments may benefit from being initialized from its 'own' assimilation cycle and we can verify our forecast results against independent (i.e. IFS) analyses.

The current operational configuration at DWD uses 90 vertical levels with a model top at $75\,\text{km}$, and a vertically nested EU domain with 60 levels and a vertical interface to the global domain at about $23\,\text{km}$. The forecast lead time is 180 h in the global domain and 120 h in the nested one. These settings are also used for our subsequent experiments unless stated otherwise. Specifically, we consider our nested EPS configuration, henceforth denoted as R2B6N7, a global $40\,\text{km}$ mesh without nest (R2B6), and a global $20\,\text{km}$ mesh without nest (R2B7). All experiment suites are conducted for January 2021, starting from the 00UTC IFS analysis for each day of the month. The physics configuration equals the operational status of autumn 2021 as far as possible, notable exceptions being that no ensemble perturbations are used and that some tuning options relying upon coupling with the operational data assimilation scheme are turned off (see Reinert et al., 2021, for further details on the ensemble perturbations). Besides the above-mentioned verification against IFS analyses, we use DWD's operational verification against SYNOP (surface) stations and radiosonde ascents (TEMP), following the WMO standard in all cases (WMO, 2019).

To exemplarily demonstrate the resolution-dependence of the forecast quality and the related nest impact, Fig. 10 shows the RMSE against IFS analyses for $500\,\text{hPa}$ geopotential and vector wind in the northern hemisphere. It is clearly evident that the errors of R2B7 are smaller than for R2B6 during the whole forecast range. The nested setup R2B6N7 exhibits slightly smaller errors than R2B6 and is, as may be expected due to the size of the refined domain, closer to R2B6 than R2B7. A closer look at Europe is given in Fig. 11, showing relative differences to R2B6 for easier readability and including an additional test in which the nest stays active until $180\,\text{h}$ (R2B6N7-180h). It is seen that during the first three forecast days, the quality improvement related to the nest feedback in R2B6N7 is comparable to what is obtained with a global $20\,\text{km}$ mesh (R2B7). Afterwards, the improvement in R2B6N7 starts to decrease as the advection from the coarse (global) domain into the nesting region becomes

more and more relevant. After five days, the errors in R2B6N7 become temporarily a bit larger than in R2B6 over Europe, and we infer from the negligible differences between R2B6N7 and R2B6N7-180h that this is unrelated to the usual termination of the nest at 120 h.

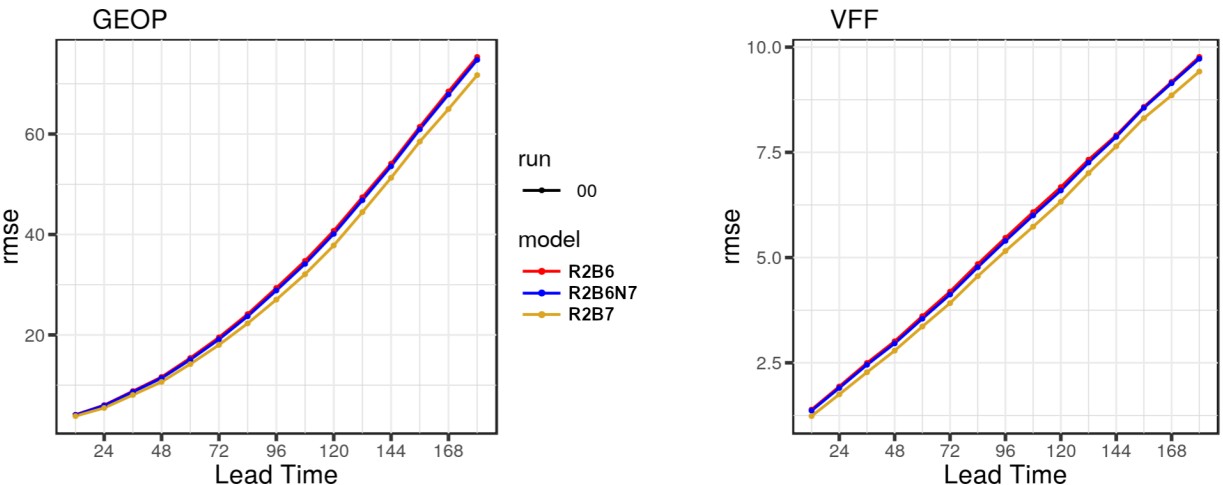

**Figure 10.** Verification against IFS analyses for the northern hemisphere (latitude $> 20°$N) at 500 hPa: RMS errors of geopotential (m) and vector wind (ms$^{-1}$) as a function of lead time (h). See text for experiment acronyms.

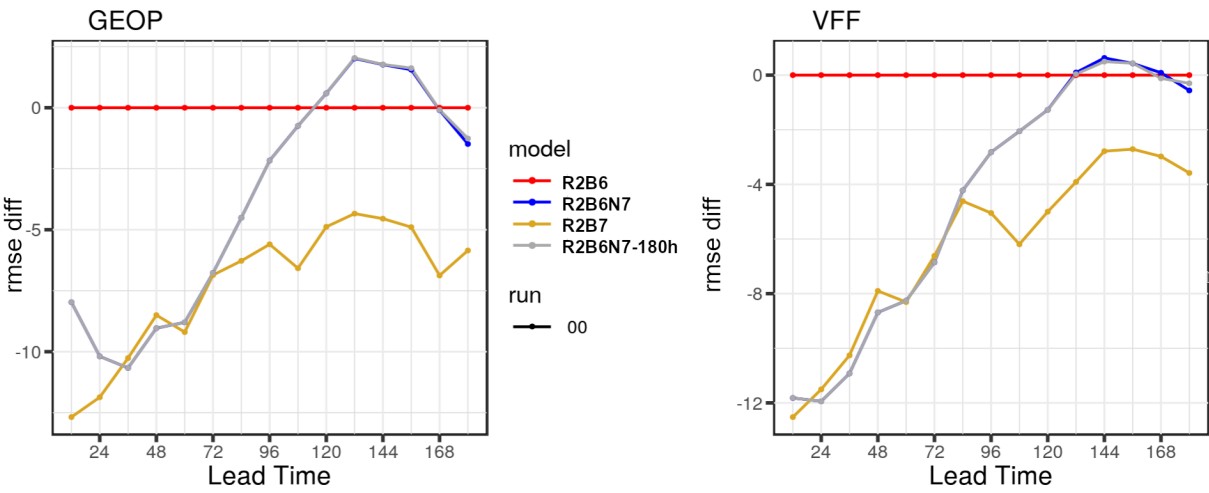

**Figure 11.** Verification against IFS analyses for Europe (35–72°N, 12°W–45°E) at 500 hPa: Relative RMS difference (%) of geopotential and vector wind w.r.t. R2B6. Note that R2B6N7 and R2B6N7-180h are identical up to a lead time of 120 h. See text for experiment acronyms.

More detailed information on the nest impact is provided by the SYNOP and TEMP verification, which include an assessment of statistical significance at the 95% level. The SYNOP verification (Fig. 12) shows a pronounced beneficial impact in

the nest overlap region (EU-NEST) during the first five forecast days, which is largest for surface pressure (PS) with an improvement of almost 10% on day 2 and still substantial for $10\,m$ wind speed (FF), $2\,m$ temperature (T2M) and $2\,m$ humidity (RH2M). After the termination of the nested domain, the differences become small and insignificant. When the nest runtime is extended to $180\,h$, the improvements continue to be present for T2M and RH2M but are negligible for PS and FF (not shown). The results for Asia show that the improvements related to the EU-nest propagate downstream with a delay of about three days. They are a bit smaller than in the nest region but still partly reach the 95% significance level, indicating an obvious benefit of our two-way nesting methodology. A more detailed view on the spatio-temporal evolution of the forecast skill improvement related to the nest feedback is provided in Fig. 13, showing a station-wise verification of surface pressure against all SYNOP stations used in the operational data assimilation cycle at DWD. On forecast days 1 and 2, the surface pressure RMSE is almost entirely smaller for R2B6N7 than R2B6, exceptions being restricted to a few individual stations. Later on, the spatial variability increases, first coherent regions with degradations appearing after about 3 days. Nevertheless, the impact of the nest stays predominantly positive, as seen in Fig. 12, and the eastward propagation of the average skill improvement is confirmed.

The TEMP verification (Fig. 14) additionally shows that the positive signal extends throughout the troposphere in both Europe and Asia. In the lower stratosphere, a slight degradation can be seen over Europe for wind speed / direction and temperature, which might be suspected to be related to the vicinity of the vertical nest interface at about $30\,hPa$. However, an additional sensitivity test with the nest extending up to $75\,km$ (not shown) revealed that this is not the case. In this test, the degradation actually extends further up into the middle stratosphere, indicating that it is probably a double-penalty effect related to resolving a larger part of the gravity-wave spectrum, an issue we generally observe when increasing the horizontal model resolution (but the discussion of which is beyond the scope of this paper). On average over the northern hemisphere, the improvements related to the EU-nest are still significant for all variables, the magnitude in the TEMP verification being largest for wind speed and direction. We finally mention that the improvement of upper-air geopotential is smaller than for PS, which appears to be a contradiction but is related to the fact that the density of radiosonde stations is more homogeneous than that of the surface stations, giving Europe an overproportional weight in the SYNOP verification. Nevertheless, this does not affect our conclusion that the regional refinement over Europe has a clear and significant benefit for the forecast quality in the northern hemisphere, demonstrating that the two-way nesting capability in ICON fully serves its purpose.

Further analysis has been undertaken to investigate potential numerical issues related to the nesting, such as possible artifacts along the lateral boundaries of the nested domain and the loss of exact mass conservation mentioned in Sect. 2. Regarding boundary artifacts, precipitation is known to be one of the most sensitive variables because unphysical flow convergences or divergences tend to induce line-shaped structures along the boundaries that are readily recognized as unrealistic. To consider this aspect, we selected a forecast run in which substantial precipitation amounts occurred along the lateral boundaries (January 15, 2021). The result after five forecast days is displayed in Fig. 15 for the nested domain and the immediate surroundings. The boundary interpolation zone, in which no prognostic precipitation is calculated in the nested domain, has been filled with interpolated values from the global domain. Although it is visible at a few spots that the interpolated precipitation along the boundaries is smoother than the prognostic one in the interior of the domain (as expected from the coarser model resolution on which it has been calculated), there is no evident discontinuity pointing to the presence of numerical problems.

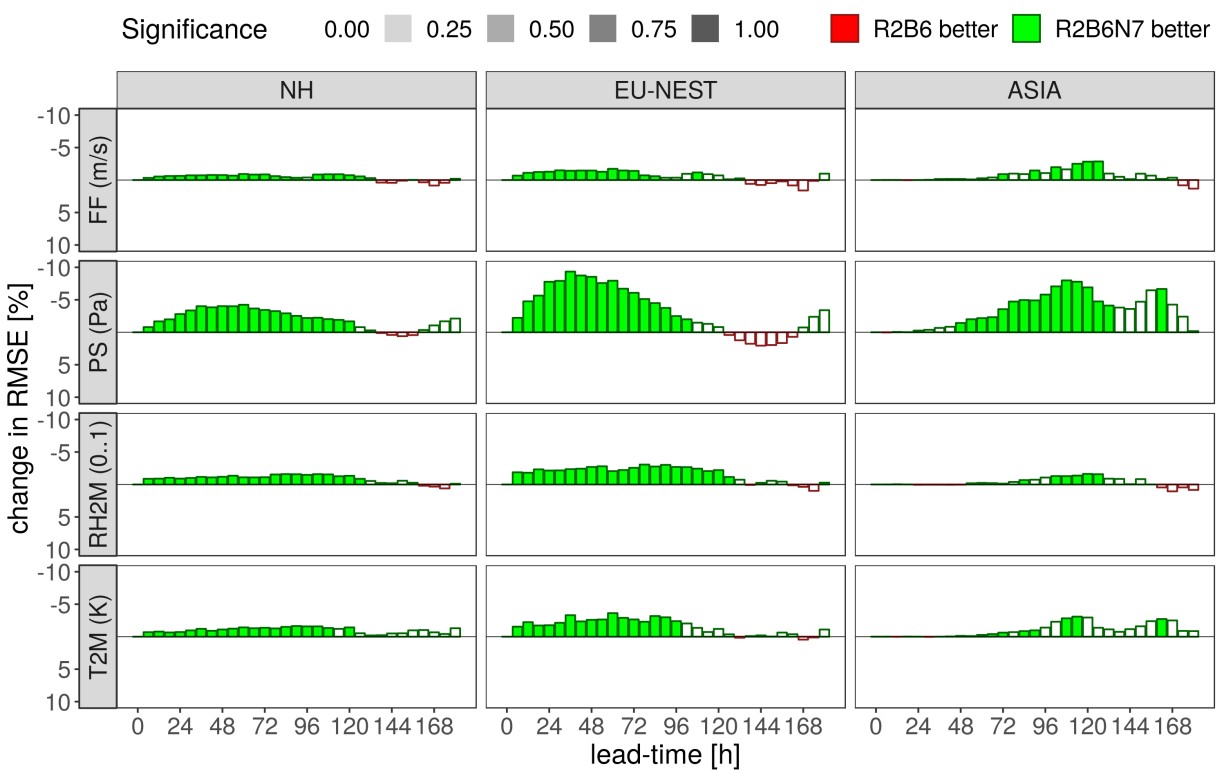

**Figure 12.** Scorecard for verification against SYNOP observations for 10 m wind speed (FF), surface pressure (PS), 2 m relative humidity (RH2M) and 2 m temperature (T2M) in the northern hemisphere (NH), the EU-nest overlap region (30–70°N, 20°W–60°E) and Asia (30–70°N, 65–140°E). Bars indicate relative RMSE differences between R2B6N7 and R2B6, and colour-filling indicates statistical significance at the 95%-level.

To examine the impact of the two-way nesting on mass conservation, an additional set of forecast experiments has been conducted in which the lead time was extended to 30 days and the nested domain stays active until the end. Results for 18 arbitrarily selected initial dates are summarized in Fig. 16, showing the relative mass change in the global domain due to
the presence of the two-way nest (the accuracy is close to machine precision otherwise). While there appears to be a slight systematic loss of mass during the first forecast days, which appears to be related to spin-up effects, there is no systematic trend visible later on. The relative errors are on the order of $10^{-6}$, which is small enough to be irrelevant for NWP purposes, and the absence of a trend indicates that the remaining conservation accuracy would even be sufficient on climate time scales (although this has not been further examined). Additional experiments indicated that the typical mass change during the first 2-3 days
is smaller and not systematically negative when starting from operational analyses (not shown). Thus, no further efforts have been undertaken to investigate the apparent spin-up effect with interpolated initial conditions.

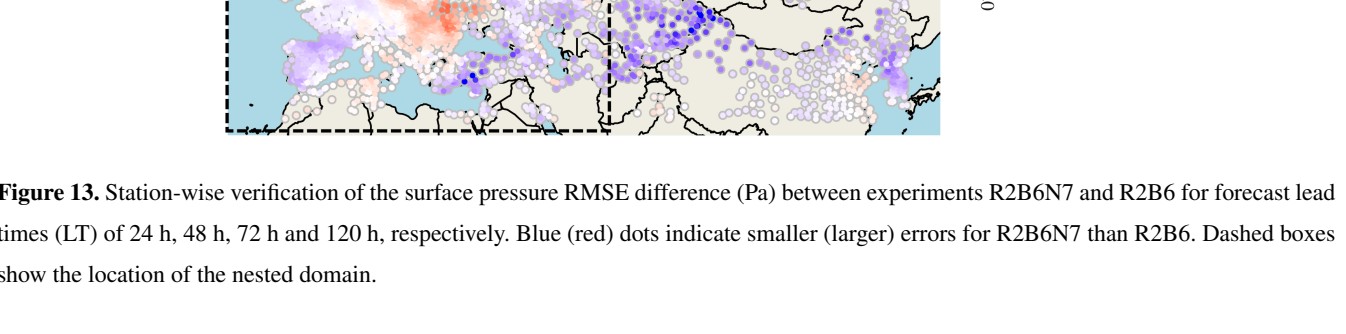

**Figure 13.** Station-wise verification of the surface pressure RMSE difference (Pa) between experiments R2B6N7 and R2B6 for forecast lead times (LT) of 24 h, 48 h, 72 h and 120 h, respectively. Blue (red) dots indicate smaller (larger) errors for R2B6N7 than R2B6. Dashed boxes show the location of the nested domain.

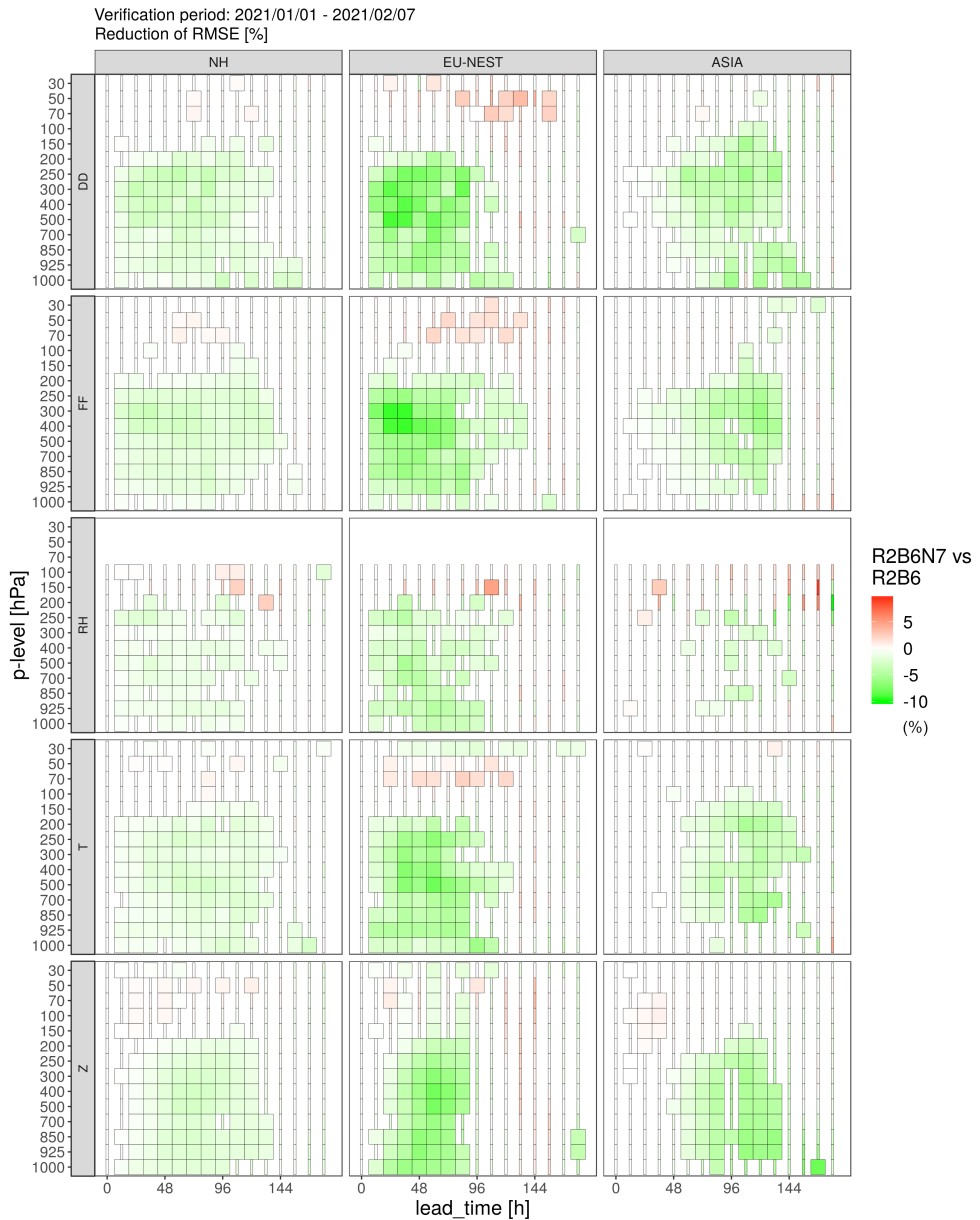

**Figure 14.** Scorecard for verification against radiosonde ascents for wind direction (DD), wind speed (FF), relative humidity (RH), temperature (T) and geopotential (Z) in the same regions as in Fig. 12. Colours indicate relative RMSE differences (%) between R2B6N7 and R2B6, and wide boxes indicates statistical significance at the 95%-level.

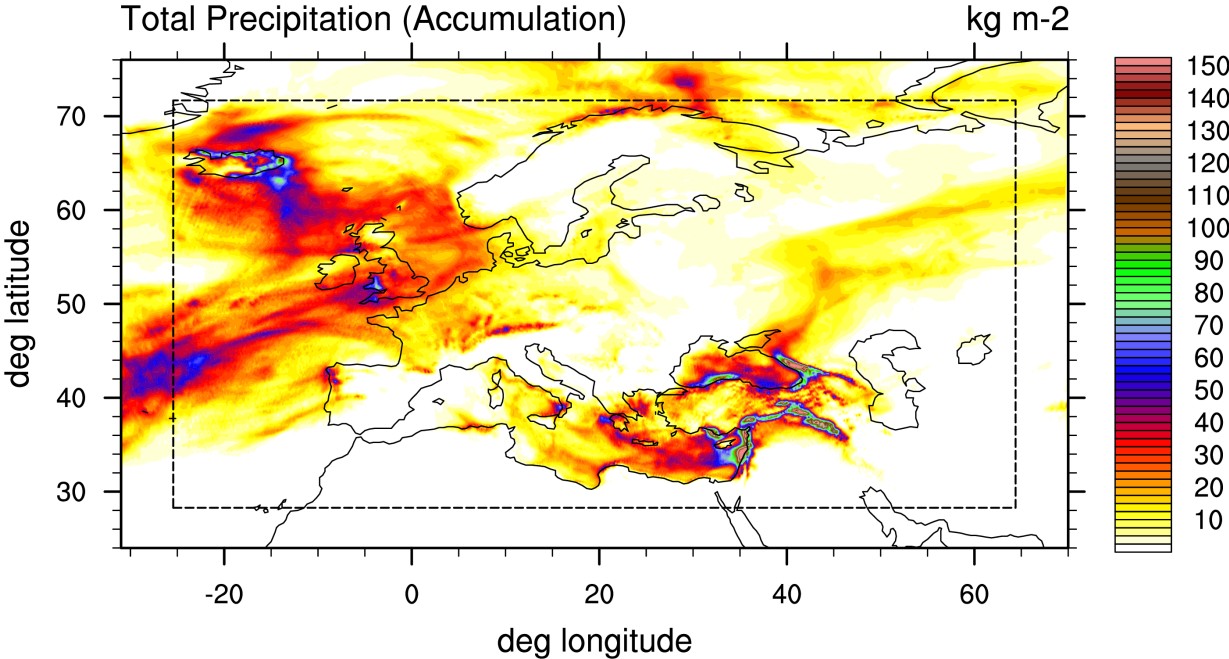

**Figure 15.** Accumulated total precipitation (kg m$^{-2}$) for a 5-day forecast starting 00 UTC, 15 January 2021. The nest outline is indicated by the dashed box. Inside (outside) the dashed box, the solution of the nested (global) domain is shown.

## 4   Conclusions

This article provides a technical description of the grid nesting implementation in the ICOsahedral Nonhydrostatic (ICON) modeling system. The available options comprise one-way and two-way nesting with one or more domains per nesting level, and vertical nesting in the sense that the upper boundary of a nested domain may be lower than that of its parent domain. In addition, a limited-area mode is available as a by-product of the grid nesting implementation, which differs from one-way nesting only in the way the lateral boundary conditions are provided. The model-internal flow control basically follows the nesting approach known from mesoscale models like MM5 (Grell et al., 1994) or WRF (Skamarock et al., 2019), but the feedback applies a Newtonian relaxation rather than a direct replacement of the prognostic variables in the overlapping part of the parent domain. The relaxation-type feedback was found to produce slightly better forecast quality in NWP applications, and obviates the need to adjust the model orography at the parent grid level. The main innovation that was needed compared to the well-established nesting implementations on quadrilateral grids was the development of appropriate interpolation algorithms for the lateral boundary conditions of the nested grids, as the usual higher-order polynomial interpolation leads to the inversion of a nearly singular matrix on the triangular ICON grid. It was found that radial basis functions constitute an adequate method. They are used either for reconstructing spatial gradients on the parent grid, followed by a linear extrapolation of the prognostic variables from the parent grid to the child grid, or for directly reconstructing the variable values on the child grid points.

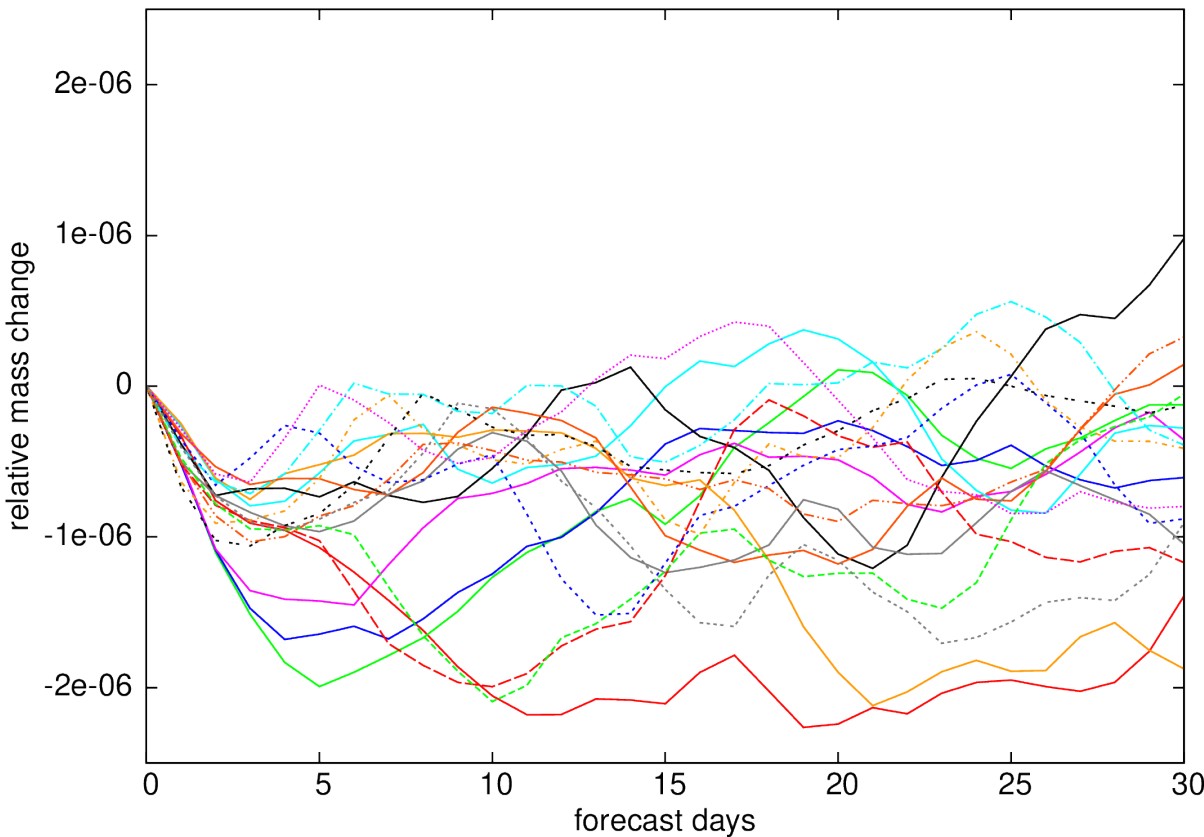

**Figure 16.** Relative mass conservation error in the global domain for 18 selected 30-day forecast experiments in January 2021 in which the nested domain remained active throughout the forecast.

To demonstrate the functionality and quality of the grid nesting in ICON, idealized tests restricted to the dry dynamical core are presented as well as real forecast experiments using a model configuration very close to that used for operational NWP at DWD. The horizontal grid nesting is addressed by two variants of the Jablonowski-Williamson test case (Jablonowski and Williamson, 2006), one considering the evolution of a baroclinic wave train triggered by a disturbance in the initial condition, and one ideally remaining in steady state in the absence of grid irregularities. Choosing the global mesh size such that the baroclinic wave is poorly resolved, the former test shows that a refined grid passed by the wave during the early stage of its development has some beneficial impact on the quality of the simulation, delivering a result lying between non-nested reference runs with the respective coarse and fine mesh sizes. Numerical disturbances related to the resolution jump along the nest boundaries are clearly smaller than the benefit of the regionally refined resolution. The steady-state variant of the test nevertheless shows that this kind of disturbances does exist and is somewhat larger than the disturbances induced by the irregularities of the uniform global icosahedral grid. The accuracy of the vertical nesting is examined with the Schär mountain test case (Schär et al., 2002), which considers a steady-state multi-scale orographic gravity wave composed of vertically

propagating and vertically decaying wave components. The results show that the wave reflections generated at the vertical nest interface are small enough to be irrelevant for practical applications. The findings from these idealized tests are corroborated by full-physics NWP experiments using the operational DWD domain configuration with a vertically nested grid over Europe and adjacent regions. Provided that the mesh sizes are chosen to lie in a range where the forecast quality has a pronounced resolution-dependence, these tests demonstrate that the feedback from the regionally refined domain to the global one has a significant beneficial impact on a variety of standard NWP scores, which also becomes evident downstream (over Asia) with a delay of about three days. We are thus able to conclude that the ICON grid nesting, which is used for operational weather forecasting at DWD since July 2015, has a practical and measurable benefit for our forecast quality.

## Appendix A: Implementation Aspects

### A1    Grid Point Indexing

In order to identify grid points belonging to certain regions of a domain and to control the feedback between parent and child domains, dedicated integer fields named `refin_ctrl` exist for cells, edges and vertices (see Fig. 1). They are provided by ICON's grid generator. Along lateral nest boundaries, these fields contain positive numbers. For cells, they indicate the shortest distance to the boundary in units of cell rows. For example, a value of 1 indicates the outermost cells. A similar counting is applied for vertices. For edges, however, the counting proceeds twice as fast, i.e. the edge-based counter is the sum of two neighboring cell counters. This fine-granuled counting enables the explicit specification of cell rows or edges up to which individual numerical operators (such as divergence, gradient or diffusion operators) are evaluated. By default, the 12 outermost cell rows (encompassing the boundary interpolation and feedback zones) are flagged by positive numbers. For the example in Fig. 1, ICON's grid generator had been configured to flag the 14 outermost cell rows, which then allows for an extended nudging zone of 10 cell rows at most.

In regions with an overlapping child domain the counters for cells, edges and vertices are filled with negative numbers (see light-blue area in Fig. 1). The counters start from $-1$ at the location of the child domain's outer boundary and descend towards the interior of the nest overlap region. The procedure is equivalent to the procedure along the lateral boundary, with the exception that it already stops after 3 cell rows. All remaining interior points are flagged with a value of $-4$ for cells and vertices and $-8$ for edges. This flagging of cells overlapping with a child domain allows to easily access all the points involved into the child-to-parent feedback process described in Sect. 2.2.2.

### A2    Distributed-Memory Parallelization

Several measures are taken in order to optimize the computational efficiency of the nesting implementation.

Model grid points in ICON are stored in (blocked) 1D vectors. In order to achieve efficient runtime flow control without the need of masking operations within loops, the grid points are ordered by their distance from the lateral boundary by making use of the `refin_ctrl` fields described in Appendix A1. Hence, grid points lying at or near the lateral boundary of a nested

domain are shifted to the beginning of the index vector. This allows excluding boundary points from prognostic computations accessing non-existing neighbor points without masking operations. In the present implementation, the four outer cell rows constituting the boundary interpolation zone (Fig. 1), and the adjacent fifth one participate in the reordering. The remaining grid points follow in a non-ordered fashion, the `refin_ctrl` indexing proceeding with the remaining gridpoints in the nudging zone, the prognostic grid points not overlapping with a child domain ($\texttt{refin\_ctrl} = 0$), and the overlapping prognostic points ($\texttt{refin\_ctrl} < 0$) (see Fig. 1). In the presence of an MPI domain decomposition, the related halo points are shifted to the very end of the index vector. Moreover, the two outermost cell rows are counted with zero weight when computing the domain decomposition, implying that the benefit of these grid points doing very little computation is not lost by load imbalance.

Regarding distributed-memory (MPI) parallelization, the default strategy adopted in ICON is to distribute all model domains among all compute processors. As this implies that child grid points are in general owned by a different processor than the corresponding parent grid point, an intermediate layer having the mesh size of the parent grid but the domain decomposition of the child grid is inserted in order to accommodate the data exchange required for boundary interpolation and feedback.

To reduce the amount of MPI communication for complex nested configurations, multiple nested domains at the same nesting level can be merged into one logical domain which is then not geometrically contiguous. This needs to be specified by the user during the grid generation process by indicating a list of individual domains that are supposed to be merged. The lateral boundary points belonging to all components of the merged domain are then collected at the beginning of the index vector. For all prognostic calculations, the multiple domains are treated as a single logical entity, and just the output files may be split according to the geometrically contiguous basic domains. As one-way and two-way nesting cannot be mixed within one logical domain, there may still need to be two logical domains on a given nest level.

To further optimize the amount of MPI communication, a so-called processor splitting is available that allows for executing several nested domains concurrently on processor subsets whose size can be determined by the user in order to minimize the ensuing load imbalance. Unlike domain merging, this allows parallel execution of one-way and two-way nested domains. This option is currently restricted to the step from the global domain to the first nesting level in order to keep the technical complexity at a manageable level.

*Code and data availability.* The ICON release version icon-2.6.4 is freely available under a personal non-commercial research license. Information on the license and instructions for downloading the code can be found at https://code.mpimet.mpg.de/projects/iconpublic/wiki/ Instructions_to_obtain_the_ICON_model_code_with_a_personal_non-commercial_research_license. By downloading the code the user accepts the license agreement. All primary data and scripts which are necessary to validate the research findings of Sect. 3.1 and 3.2 are freely available for download from the Open Research Data Repository of the Max Planck Society (Edmond) under https://doi.org/10.17617/3. NOC2AE (Zängl et al., 2022). Data and scripts for the NWP applications of Sect. 3.3 are not included due to the huge size of the data set (about $3.3\,\mathrm{TB}$), and the fact that installing and running the verification toolchain outside DWD's computer systems is beyond the authors' expertise. Access to these data will be granted upon request (contact: guenther.zaengl@dwd.de).

*Author contributions.* All authors contributed to the writing of this paper. GZ conducted the initial nesting implementation and ran the experiments of Sect. 3.1 and Sect. 3.3. DR worked on the vertical nesting and ran the experiments of Sect. 3.2. FP developed the grid generator, and worked on parallelization aspects and performance optimizations.

*Competing interests.* The authors declare that they have no conflict of interests.

*Acknowledgements.* The authors would like to thank the entire ICON team for their excellent work, without which this would not have been possible. Our additional thanks go to Felix Fundel (DWD) who has developed the verification toolchain used in this work. The authors also appreciate the constructive comments made by the two anonymous reviewers, which led to a significant improvement of the manuscript.

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
