# Peer review of "Grid Refinement in ICON v2.6.4"

_Geoscientific Model Development, 2022_

## Referee Comment (RC2)

Review of *Grid Refinement in ICON v2.6.4*
by Zängl, Reinert and Prill
June 2022

The authors present 1- and 2-way grid nesting algorithms for the ICON model. Grid nesting has a long history in atmospheric modeling and the authors focus their attention on describing differences from existing implementations in other models, with these differences arising primarily from the authors use of a triangular primal grid in ICON compared to the rectangular grid implementations referenced by the authors. The implementation is comprehensive and reasonably complete. The simulation test results support the algorithmic choices and suggest that the implementation is correct. The points I raise below are primarily for clarification and for missing information.

1. Lines 28-30: Implementing nesting with a hexagonal C-grid is not more difficult, for example see the regional version of MPAS (Skamarock et al 2018, doi:10.1175/MWR-D-18-0155.1), where the basic machinery exists. 2-way nesting would follow from applying the techniques described in Dubos and Kevlahan (2013, DOI:10.1002/qj.2097) which demonstrate and an adaptive mesh shallow-water model using a hexagonal C-grid.

2. Line 110: "*the wish for consistency with continuity*" does not necessitate coupling of the dynamics on the substeps $\Delta\tau$. Consistency for tracer transport only requires that the time-averaged mass fluxes (averaged over the substeps) on the cell faces are used for this transport.

3. Line 113: "*multi-grid approach*": "Multi-grid" could easily be confused with "multigrid" methods for solving PDEs. I suggest the authors recast this as, for example, *The static mesh refinement in ICON is accomplished using multiple separate grids.*

4. Line 118: "*domains having the same parent are not allowed to overlap*": Why?

5. Lines 118-119: Is the nesting configuration a compile-time specification, or a run-time specification?

6. Line 125: Why is the refinement ratio fixed at a factor of 2? Triangles can be divided by any integer division of the edges.

7. Lines 138-140, and Figure 1: It appears that the use of triangles for the primal grid results in the need for large number of cells in the boundary interpolation zone. Nested grid implementation on rectangular grids (MM5, WRF, and FV3 references) have far fewer cells in their boundary interpolation zone (also called the specified zone), as does the 1-way nesting used in the regional version of MPAS (2018, doi:10.1175/MWR-D-18-0155.1) that uses the dual of the ICON grid (hexagons). This results in more interpolations, and this should be noted.

8. Line 157: If vertical nesting is not applied, is using relaxation instead of replacement beneficial?

9. Figure 1: I was confused by this figure at first. In the light-blue region the fine-grid cells are explicitly drawn. At first I had thought that the fine grid contained coarse-grid cells in the nudging zone and boundary interpolation zone, but after reading much further in the paper I understood this is not the case. The indexing of the fine-grid edges with white lettering is not readily apparent.

10. Figure 1, figure caption: "Another child domain overlapping with the depicted domain." I do not understand this sentence because earlier the authors stated that domains cannot

overlap (comment 3).  Do the4 authors wish to state this is where the child and parent domains are coincident?

11. Lines 280-285:  Using the interpolation constraints (5), (6) and (7), for perfect triangles $\alpha_j = \frac{1}{4}$ for all j, and this would be identical to an area-weighted computation for all the weights (i.e. (7) for all j). Is the problematic checkerboard pattern produced because of the necessarily non-perfect triangles in the spherical grid?

12. Section 2.4 (line 390): Is this processing sequence any different from that in WRF or FV3?  I do not see any difference from the process used in WRF.

13. Figure 4: Why is only the parent grid solution plotted in this figure?  I expected that the combined solutions would be plotted, where the highest resolution data is used in any region of the plot.  I appreciate that the nest values are plotted in figure 5, but separating them makes it difficult to examine the continuity of the solution across the nest boundaries.

14. Lines 380-385: How is the vertical flux for scalars computed without a specified zone extending vertically away from the upper nest boundary?  I appreciate how the boundary fluxes are computed, but is it the case that the flux one interface level down doesn't require values above the upper boundary?

15. How does the vertically-implicit solver in the dynamics handle an upper boundary where *w* is non-zero?  Perhaps stated differently, is specifying (interpolating from the parent grid) the vertical velocity at the upper boundary sufficient?

16. Figure 6:  The solution plotted here for the non-nested experiment (6a) appears to be significantly better than the ICON solution plotted in figures 8 and 10 in Lauritzen et al 2010 (doi:10.3894/JAMES.2010.2.15), where a number of models were compared using this test case.  If this is the case then the authors may want to point that out.  Also, results from this case are usually presented for day 9, so the authors' day 10 plots make comparison more difficult.

17. Figure 10: The vertical axis is mislabeled.  It is an "RSME difference" and not an RSME. It also seems to me it would be more natural to compute it is a difference from R2B7 which is effectively the reference (higher-resolution) global solution in this case. By using R2B7, the difference (i.e., relative error) would be growing over time.

18. Section 3.2: Specific mention of this test being performed on a reduced-radius sphere might be helpful to readers unfamiliar with this version of the test.

19. Section 3.2: Should the reader understand that there is no Cartesian-plane perfect triangle version of ICON available to do idealized 2D and 3D tests?

20. Section 3.2: While the results show little issue with the upper boundary formulation in the nested simulation, the results are difficult to compare with the previous studies cited by the authors (Zängl et al 2015; Skamarock et al 2012).  The nest upper boundary is at 20 km, but the Zängl et al results are only shown up to 12 km, and the Skamarock et al results are only given through 10 km and are on computed 2D Cartesian x-z plane as opposed to a sphere.  Klemp et al 2015 (https://doi.org/10.1002/2015MS000435) show solutions on the sphere for this test case (although using a slightly different y variation of the mountain amplitude), again only through 10 km in height.  Perhaps a grid can be constructed such that the upper boundary could be placed at 12 km and plots exactly like those in Zängl et al (2015) for easy comparison?

21. Appendix A1: The decision to reorder the grid points is an interesting one.  Other models employing unstructured grids, for example Skamarock et al (2018) regional MPAS uses

masks and report low overhead associated with their usage. Is the decision to re-order in part driven by the use of triangles for the primal grid with the resulting large number of cells in the interpolation zone?

22. Appendix A2: Line 647-648: "*multiple nested domains at the same nesting level can be merged*". Does this mean the default is not to merge them?

23. Appendix A2: The WRF model distributes each domain over all the processors. This appears to be the default for ICON. Is it the case that the newer features regarding distributed memory configuration are the option of merging multiple domains on a given level, and the processor splitting described in lines 664-667?

---

## Author Comment (AC1)

**GMD-2022-120: Response to reviewer comments**

We would like to thank the two anonymous reviewers for their constructive comments, which led to a significant improvement of our manuscript.

**Reply to Reviewer 1**

L8-10: This is a little too technical to write on the abstract. The relevant description is not written in the main part of this paper but in Appendix. As another comment from a different point of view, this description appears some techniques of vectorization. In my understanding, this is of course important for DWD system as the center uses the NECs supercomputers. However, these techniques may not be generally effective to other supercomputers.

We agree that this technical aspect does not need to be highlighted in the abstract, and we removed the sentence from the abstract. Regarding the platform-dependence of this optimization measure, we'd like to mention (without providing a discussion in the manuscript) that the implementation was made several years before DWD got its current NEC supercomputer. Actually, vectorization on our previous Cray XC30/XC40 was much more sensitive to IF masks than it is on our present NEC.

L123: "or convection can be reduced by stronger entrainment..."... I understand that enhancement of entrainment, particularly for deep convection, can weaken the effects of convective parameterization. However, it does not mean reducing the convection itself. This description could mislead readers who are not familiar with convective parameterization. I suggest or effects of convective parameterization can be weakened by enhancement of entrainment for deep convection as alternative.

We clarified the formulation into "or the convection scheme can be tuned differently or even be switched off completely"

**L224-225: It should described that how much is the magnitude of the diffusive coefficient is and how the magnitude was determined.**

More detailed information on the boundary diffusion has been added to the manuscript: For the secondorder diffusion, a coefficient of  $0.005 a_e/\Delta t$  is applied, where  $a_e$  is the area represented by the current triangle edge, and the scaling with  $\Delta t$  means that the amount of diffusion is effectively independent from the time step. This proved to be sufficient to suppress the development of spurious disturbances even in the data assimilation cycle, which turned out to be the most critical application mode in this respect.

L280: I agree with that the feedback operator does not guarantee the mass conservation, as mentioned in this line. In a pragmatic point of view, how much the feedback operator loses the mass conservation could interests readers. If authors put some figures to respond to the interests, this manuscript could be improved. For examples, comparison of total mass of the child domain between parent model with/without two-way nest, and child model as a function of forecast lead time in JW test (Section 3) and/or NWP case studies (Section 4)

To investigate this aspect, we conducted an additional set of NWP forecast experiments with extended lead time (30 days) and the nest staying active throughout the forecasts. The results are reported at the end of section 3.3. We decided to show the relative mass change in the global domain, for which the ideal result is obvious. The mass in the nested domain would not be well suited for this purpose because it is not a conserved quantity. It strongly fluctuates related to the synoptic-scale evolution, and differences in the child domain mass between 1-way and 2-way nesting predominantly reflect the impact of the feedback on the synoptic evolution, rather than highlighting local conservation issues.

L388: The multi-nesting using the recursive approach sounds interesting, however, the rest of this paper

**does not present any results from the multi-nesting experiments. Section 2.4 should be shortened and moved to Section 4 as future work, or should be moved to Appendix.**

In Section 2.4, we pointed out more clearly that multi-domain same-level and multi-level nesting is not an envisaged feature for the future, but that this feature is already implemented and used by the ICON community. What is, however, missing so far is a comprehensive technical description of this feature, which is one focus of this paper. An example of multi-domain same-level nesting is already given as part of the baroclinic wave tests in Sect. 3.1 (see Fig. 4e,f). Regarding multi-domain multi-level nesting we added a recent reference, where mountain-wave induced PSCs are simulated using a global ICON(-ART) configuration with three successively nested domains (Weimer et al.,2021).

L437-440: It is difficult to see the difference in phases from Figure 4. Difference between E2 or E3 and E1 should be shown.

We agree that the differences in phase lag are in parts hard to see. To ease the comparison, we added dashed gray lines to all surface pressure plots, indicating the longitudinal position of the middle and trailing cyclones' minimum surface pressure in the R2B5 reference run (E5).

L440-442: It is difficult to see the difference in phases from Figure 4. Difference between E2 and E3 should be shown.

In this case, it is our intention to show that the difference is negligibly small, as pointed out in the related discussion.

L467: For readers who are not familiar with icosahedral grids, some references are necessary to describe "well known regular wavenumber-five disturbance pattern characteristic for icosahedral grids"

Thank you for this hint. We have added two references which discuss the grid imprinting for various types of grids on the basis of the JW test (icosahedral, cubed-sphere, etc.).

L471-473: It is natural that the nested run has the larger errors than those in the non-nested run since the nested run has more source of numerical disturbances. The authors should explain the reason why the disturbances due to the boundary are not problematic in fine meth models or NWP cases. For examples, the authors could emphasize following points more clearly.

(1) the JWs steady state test is initialized from a baroclinically unstable basic state, hence the result is very sensitive to small perturbations. The basic state in the JWs experiment is highly idealized to extract models characteristics.

(2) disturbance error due to the nesting is smaller as the meshes are finer.

(4) These are the reasons why kinds of errors in Figure 6 are not obvious in fine mesh models or NWP cases

We clarified the discussion of the practical relevance of the numerical disturbances encountered in the steady-state JW test, now mentioning explicitly that the real atmosphere is not as baroclinically unstable as the JW initial condition because the instability is continuously depleted by synoptic-scale disturbances.

L501-502: It is not clear that Spurious disturbances means Spurious disturbances against the analytic solution or Spurious disturbances against the reference. the uppermost quasi-hydrostatic wave crest and trough, and to the leeward propagating wave signal are also seen around the region where x > 20km in the reference (Figure 7 (a))

We clarified the formulation into "deviations from the reference result".

L511-517: I am not sure that

(1) whether the parent model is more consistent with the reference than the model without vertical nesting. Because the result from the model without vertical nesting is not shown.

(2) whether the more consistent result with the reference becomes closer to the truth (the analytic solution).

Because the leeward wave propagation is not obviously seen in the analytic solution (e.g. Fig (1) of Klemp (2003), MWR, https://doi.org/10.1175/1520-0493(2003)131<1229:NCOMTI>2.0.CO;2)

(1) We clarified that the non-nested reference result is shown in Fig. 12a.

(2) We also clarified that the variable Brunt-Väisälä-frequency setting used in our study in order to allow for the desired model top of 40 km (without getting negative temperatures) does not allow for comparing against an analytic solution. However, this type of validation is not attempted here because model results serving this purpose have already been published by Zängl et al. (2015).

L519 (general comment on Section 4): From experience of nested runs of regional models, errors due to boundary conditions may be visualized by hydrological parameters such as clouds, precipitations, etc. If the authors attempt to present that spurious disturbances are not clearly seen, showing some figures of forecast snapshots from the parent model could be necessary.

To address this aspect, we added a figure showing accumulated precipitation over 5 days for a forecast run with sufficiently strong precipitation along the nest boundaries. The discussion is given in section 3.3. Briefly summarized, the figure confirms that numerical disturbances along nest boundaries are small in ICON.

L561: It is very interesting that impacts of EU-nesting propagate downstream with a delay over the ASIA region. It could be more persuasive if the authors can show the propagation of error diff. on the maps. (e.g. RMSE diff. of sea level pressure on the map at day1, day2, ..., and day5 etc.)

We added another figure to section 3.3 showing the average RMSE difference of surface pressure on a map including all SYNOP stations available in the relevant region. It clearly shows how the error reduction related to the nest feedback evolves with time and propagates eastward.

L610: I suggest that the authors write down possible future works or plans of this study in the end of Section 4. Propagating the impacts of the two-way nesting downstream is the interesting and gives important implication. This finding implies the possibility of another nested domain upstream from the EU region for improving medium-range forecast over the EU region. This point could be a future prospect of this study in addition to multi-ways nesting described in 2.4.

We feel that discussing upcoming upgrades of our operational system would be beyond the scope of this paper, the primary purpose of which is to document the nesting implementation in ICON. Actually, we are preparing a horizontal resolution upgrade from 40/20 km to 26/13 km in the ensemble part of our system, and a vertical resolution upgrade from 90 to 120 levels in the deterministic as well as the ensemble part. We also considered extending the nested domain over the Atlantic towards the American coast, but the cost-benefit ratio of this change turned out to be not as good as for the general resolution increase. This is consistent with the widespread experience that the benefit of higher model resolution comes to a large extent from better resolving the orography, implying that the impact is more pronounced over land than over oceans.

L13: The order of description should be consistent with that in Section 3 (3.1 for JW tests and 3.2 for Schars mountain wave tests)

The order has been corrected.

Figure 1: White integers which indicate the indexing edged are a little bit difficult to read. For an example, black characters in white circles could make the figure more reader- friendly.

Figure 1 and the figure caption have been revised in order to improve readability.

**Reply to Reviewer 2**

1. Lines 28-30: Implementing nesting with a hexagonal C-grid is not more difficult, for example see the regional version of MPAS (Skamarock et al 2018, doi:10.1175/MWR-D-18-0155.1), where the basic machinery exists. 2-way nesting would follow from applying the techniques described in Dubos and Kevlahan (2013, DOI:10.1002/qj.2097) which demonstrate and an adaptive mesh shallow-water model using a hexagonal C-grid.

We clarified the main reason why we regard a triangular grid as more suitable for a two-way nesting capability than a hexagonal one: Triangular grids of successive refinement levels have a unique relationship between parent and child cells, whereas the majority of hexagonal child cells is shared between two adjacent parent cells. We are not saying that it is impossible to develop viable alternatives for hexagonal grids, but it appears quite clear that the method developed by Dubos and Kevlahan is more complex and mathematically more demanding than a conventional two-way nesting.

2. Line 110: the wish for consistency with continuity does not necessitate coupling of the dynamics on the substeps Dt. Consistency for tracer transport only requires that the time- averaged mass fluxes (averaged over the substeps) on the cell faces are used for this transport.

We fully agree with the reviewer. The key requirement for consistency with continuity is that the mass fluxes 'seen' by the air mass continuity equation and the tracer mass continuity equations are the same. In this sense, our initial statement was misleading as it might give the impression that consistency with continuity in the presence of nested domains is automatically achieved by coupling at the fast physics timestep, irrespective of the way mass fluxes are specified.

Our intention was to point out that, in the presence of nests, there are additional constraints that need to be fulfilled in order to maintain consistency with continuity. Consistency with continuity can easily be destroyed in an otherwise consistent model, when adding a two-way nesting without care.

We revised our formulation and point out more clearly what we think are the additional constraints. Firstly the mass fluxes specified at nest lateral boundaries must be consistent (in addition to the interior fluxes) in the sense that the air mass and tracer mass continuity equations 'see' the same mass fluxes. Otherwise consistency with continuity on the child domain is violated. Secondly, for two-way nesting the childto-parent feedback increments for tracer masses must sum up to the feedback increment for air mass. Otherwise consistency with continuity is violated on the parent domain. Both aspects can only be fulfilled in a natural way when coupling the nested domains at the fast physics timestep  $\Delta t$ , such that the parent mass fluxes are available at the required time level when executing the boundary interpolation.

3. Line 113: multi-grid approach: Multi-grid could easily be confused with multigrid methods for solving PDEs. I suggest the authors recast this as, for example, The static mesh refinement in ICON is accomplished using multiple separate grids.

The formulation has been modified accordingly.

4. Line 118: domains having the same parent are not allowed to overlap: Why?

We clarified that we mean that multiple child domains at the same nest level are not allowed to share the same parent grid cells because this would lead to ambiguities in combination with two-way nesting.

5. Lines 118-119: Is the nesting configuration a compile-time specification, or a run-time specification?

We clarified that the domain configuration is controlled by the grid files provided as input (and thus by namelist settings).

6. Line 125: Why is the refinement ratio fixed at a factor of 2? Triangles can be divided by any integer division of the edges.

We clarified that this decision was made because higher nesting ratios increase the probability for numerical artifacts along nest boundaries (e.g. by partial wave reflections).

7. Lines 138-140, and Figure 1: It appears that the use of triangles for the primal grid results in the need for large number of cells in the boundary interpolation zone. Nested grid implementation on rectangular grids (MM5, WRF, and FV3 references) have far fewer cells in their boundary interpolation zone (also called the specified zone), as does the 1-way nesting used in the regional version of MPAS (2018, doi:10.1175/MWR-D-18-0155.1) that uses the dual of the ICON grid (hexagons). This results in more interpolations, and this should be noted.

We added a sentence explaining that halving the size of the boundary interpolation zone (to the minimum needed due to the domain decomposition constraints with two-way nesting) would be possible with some modifications to wide stencil operations near boundaries, (e.g. reducing  $\nabla^4 v_n$  to  $\nabla^2 v_n$ ) but this effort has never been made because there would be no benefit for computational efficiency. As mentioned in the revised appendix B, the outer boundary points are not counted when computing the domain decomposition, so that they do not contribute to the total computing time by inducing load imbalance.

8. Line 157: If vertical nesting is not applied, is using relaxation instead of replacement beneficial?

As mentioned in the revised version, the quality difference between direct feedback and relaxation feedback is small without vertical nesting.

9. Figure 1: I was confused by this figure at first. In the light-blue region the fine-grid cells are explicitly drawn. At first I had thought that the fine grid contained coarse-grid cells in the nudging zone and boundary interpolation zone, but after reading much further in the paper I understood this is not the case. The indexing of the fine-grid edges with white lettering is not readily apparent.

The Figure has been revised and the Figure caption has been slightly rephrased. I.e. we point out that the depicted nest corresponds to domain 2 shown in the schematic in the upper left, and that the fine grid depicted in the light blue region corresponds to domain 3, which is nested into domain 2.

10. Figure 1, figure caption: Another child domain overlapping with the depicted domain. I do not understand this sentence because earlier the authors stated that domains cannot overlap (comment 3). Do the authors wish to state this is where the child and parent domains are coincident?

The formulation has been clarified; we mean another child domain at the next-higher nesting level.

11. Lines 280-285: Using the interpolation constraints (5), (6) and (7), for perfect triangles  $a_j = 1/4$  for all *j*, and this would be identical to an area-weighted computation for all the weights (i.e. (7) for all *j*). Is the problematic checkerboard pattern produced because of the necessarily non-perfect triangles in the spherical grid?

Yes, the distortion of real spherical triangles is exactly the reason for the checkerboard issue. We added a sentence pointing this out.

12. Section 2.4 (line 390): Is this processing sequence any different from that in WRF or FV3? I do not see any difference from the process used in WRF.

Yes, the processing sequence is probably the same as in WRF. However, as this is a vital part of ICON's nesting capability, we feel that adding a short description of this processing sequence is nevertheless desirable in order to make the description complete. We added a sentence at the end of Sect. 2.4 pointing out that the processing sequence is very similar to WRF and FV3. At the same time, we pointed out that notable differences still exist at a more detailed level. According to Mouallem et al. 2022, the child to parent update (feedback) in FV3 is restricted to temperature and the wind components, which appears to require executing the feedback before the physics call at the parent level in order to maintain numerical stability. In ICON, on the other hand, density and non-precipitating moisture tracers are included in the feedback,

and the feedback is executed after completing the parent-level time step (including physics updates).

13. Figure 4: Why is only the parent grid solution plotted in this figure? I expected that the combined solutions would be plotted, where the highest resolution data is used in any region of the plot. I appreciate that the nest values are plotted in figure 5, but separating them makes it difficult to examine the continuity of the solution across the nest boundaries.

In this set of experiments, our main focus was on how the nested domains impact the solution on the parent domain via the incremental feedback mechanism. Any solution discontinuities or noise that might develop in the nested domain is likely to become evident in the parent domain due to the two-way coupling. However, we admit that this is a rather indirect way of checking for solution discontinuities or noise, as part of the unwanted flow structures might be filtered by the fine-to-coarse grid interpolation of the feedback increments as well as the comparatively large feedback timescale.

In order to point out more clearly that we do not encounter significant solution discontinuities or accumulation of noise along the nest boundaries, we added Fig. 6, which complements Fig. 4(g,h) and shows the combined solution (i.e. nest solution with the parent solution outside of the nest region). In order to give the reader an idea about the location of the boundary interpolation zone in this experiment (i.e. which part of the nest is prognostic and which is filled with values interpolated from the parent domain), we added a white-shaded region to Fig. 6(a), which indicates all nest boundary points.

14. Lines 380-385: How is the vertical flux for scalars computed without a specified zone extending vertically away from the upper nest boundary? I appreciate how the boundary fluxes are computed, but is it the case that the flux one interface level down doesnt require values above the upper boundary?

Computing the flux boundary condition for scalars at the nest interface level does not require any vertical boundary interpolation (or specified) zone, as the boundary conditions are entirely derived by horizontal parent to child interpolation. The need for vertical interpolation is avoided in the current model version due to the fact that the vertical level distribution is the same for the parent and child domain and the constraint that the nest interface level must be located at a height, where the (parent and child) coordinate surfaces are flat.

However, the reviewer is right that one might think of situations in which a specified zone for tracer mass fractions  $q_k$  above the upper boundary is required in order to compute the vertical tracer mass flux one interface level down. Such situations might occur in regions with very strong vertical downdrafts where the local Courant number (CFL) based on the fast physics time step  $\Delta t$  exceeds 1. Our current implementation does not account for these situations, i.e. the flux computation in the first layer below the nest interface is only stable for CFL  $\leq 1$ . We note, however, that in practice the nest upper boundary is usually located somewhere in the lower stratosphere or even higher, where the vertical layer thickness is usually sufficiently large to stay within the CFL limit.

Nevertheless we thank the reviewer for pointing this out. In Sect. 2.4 we have added a comment stating that the vertical tracer advection is limited to CFL < 1 in the layer below the nest upper boundary.

15. How does the vertically-implicit solver in the dynamics handle an upper boundary where w is nonzero? Perhaps stated differently, is specifying (interpolating from the parent grid) the vertical velocity at the upper boundary sufficient?

Yes, the reviewer is right. For the vertically implicit solver, i.e. in order to close the tridiagonal system for w, it is sufficient to specify the vertical velocity at the upper boundary. Without vertical nesting (and if a rigid lid is assumed), we explicitly specify  $w = 0 \text{ ms}^{-1}$  at the upper boundary, whereas for vertically nested domains we derive w by horizontal parent to child interpolation. We added a short comment to Sect. 2.4 to clarify that the boundary condition for w is required for the vertically implicit solver, whereas the mass flux  $\rho w$  is required for the vertical divergence terms in the prognostic equations for  $\rho$ ,  $\pi$  and  $\rho q_k$ .

16. Figure 6: The solution plotted here for the non-nested experiment (6a) appears to be significantly better than the ICON solution plotted in figures 8 and 10 in Lauritzen et al 2010 (doi: 10.3894/JAMES.2010.2.15), where a number of models were compared using this test case. If this is the case then the authors may want to point that out. Also, results from this case are usually presented for day 9, so the authors day 10 plots make comparison more difficult.

Thank you very much for this hint. Indeed, the result of the first DCMIP study (Lauritzen et al 2010) was obtained with an early version of the hydrostatic dynamical core (Wan et al 2013), which did not perform as well as the final nonhydrostatic dycore. We added a short remark in Sect. 3.1.

To simplify comparison with other models we replaced the results presented for day 10 by results for day 9.

17. Figure 10: The vertical axis is mislabeled. It is an RSME difference and not an RSME. It also seems to me it would be more natural to compute it is a difference from R2B7 which is effectively the reference (higher-resolution) global solution in this case. By using R2B7, the difference (i.e., relative error) would be growing over time.

We corrected the label of the vertical axis. However, to be consistent with the subsequent figures, we prefer retaining R2B6 as the reference for the difference plots. Our main goal is to highlight the improvements related to the nested domain, and the additional reference to the R2B7 result just serves to demonstrate that the R2B6N7 scores are close to R2B7 during the first forecast days in the nest overlap region.

18. Section 3.2: Specific mention of this test being performed on a reduced-radius sphere might be helpful to readers unfamiliar with this version of the test.

We added a footnote declaring that we are not using the DCMIP small planet variant of the Schär test, but the original one with the differences stated in the manuscript.

19. Section 3.2: Should the reader understand that there is no Cartesian-plane perfect triangle version of ICON available to do idealized 2D and 3D tests?

ICON actually has an option for a planar double-periodic torus grid, but this is option is not prepared for nesting.

20. Section 3.2: While the results show little issue with the upper boundary formulation in the nested simulation, the results are difficult to compare with the previous studies cited by the authors (Zängl et al 2015; Skamarock et al 2012). The nest upper boundary is at 20 km, but the Zängl et al results are only shown up to 12 km, and the Skamarock et al results are only given through 10 km and are on computed 2D Cartesian x-z plane as opposed to a sphere. Klemp et al 2015 (https://doi.org/10.1002/2015MS000435) show solutions on the sphere for this test case (although using a slightly different y variation of the mountain amplitude), again only through 10 km in height. Perhaps a grid can be constructed such that the upper boundary could be placed at 12 km and plots exactly like those in Zängl et al (2015) for easy comparison?

We clarified that the primary purpose of this test and its specific configuration is testing the functionality of the vertical nesting. As mentioned in lines 353/354 of the original manuscript, vertical nesting is supported only at heights where the coordinate surfaces are exactly flat, which requires a nest interface above 15 km and thus a relatively high model top (40 km), which in turn necessitates a higher Brunt-Väisälä frequency in the stratosphere in order to avoid negative temperatures. We are aware that this precludes a strict comparison with analytic solutions and previously published results, but this should be acceptable because ICON results serving this purpose have already been presented by Zängl et al. (2015).

21. Appendix A1: The decision to reorder the grid points is an interesting one. Other models employing unstructured grids, for example Skamarock et al (2018) regional MPAS uses masks and report low overhead associated with their usage. Is the decision to re-order in part driven by the use of triangles for the primal

**grid with the resulting large number of cells in the interpolation zone?**

The original motivation for the reordering was given by the fact that the unstructured grid allows doing it, combined with the desire to avoid IF conditions at many places of the dynamical core which otherwise would have been needed to avoid accesses to non-existing boundary grid points. Later on, we realized that the strong scaling properties of the model code can be improved by moving the MPI halo points to the end of the index vector. As mentioned in the reply to comment 7, the primary compensation for the large boundary interpolation zone comes from not counting the two outer rows when computing the domain decomposition. Thereby, the fact that very few computations are done on these grid points does not contribute to load imbalance.

**22. Appendix A2: Line 647-648: multiple nested domains at the same nesting level can be merged. Does this mean the default is not to merge them?**

Yes, we further clarified that the domain merging needs to be requested actively by the user. One motivation for the default being not to merge is that many postprocessing applications expect contiguous domains.

23. Appendix A2: The WRF model distributes each domain over all the processors. This appears to be the default for ICON. Is it the case that the newer features regarding distributed memory configuration are the option of merging multiple domains on a given level, and the processor splitting described in lines 664-667?

Without giving a specific discussion in the paper, we'd like to mention here that the processor splitting in ICON has already been implemented about 10 years ago. At that time, it was planned to use a very complex configuration with a mixture of one-way and two-way nested domains for operational production, but this plan was abandoned before starting the operational use of ICON in January 2015. The runtime optimizations implemented for this purpose have been retained afterwards even though they are very rarely used in practice.

---

## Referee Report (RR1)

I would like to thank to the authors for revising and improving the manuscript. The revised manuscript has become much more persuasive to show the benefit of the suggested grid refinement method.

Most of my concerns have been resolved. I think that the revised manuscript is worth publishing after minor revisions (e.g grammatical correction, revising figures for readability etc).

* L510: "we note that the baroclinic instability of the real atmosphere is much smaller than in the initial state of the JW test..."

   -> "instability is smaller " appears to be grammatically strange.

   For an example, "we note that the disturbance growth due to the baroclinic instability in the real atmosphere is much weaker than in the initial state of the JW test..."

   could be an alternative. (However, I am not an English native speaker, so, I leave it to the authors and editor's decision.)

*L563: "On the other hand, this test has shown the ability of the child-to-parent feedback mechanism to have a small but noticeable positive impact on the quality of the nest interface conditions."

   -> I interpret that "positive impact on the quality" is corresponding to representing leeward propagation in the R2B11 parent domain more consistently with the R2B12 reference run, than that in the R2B11 without nesting. If my understanding is correct, the author could write down that more explicitly around L560.

*In Figures 13 and, 15, which are newly added to the revised manuscript, boxes showing the nested domain or boundaries of the interpolation zone could be plotted for readability, as in Figures 4, 6, and 7.
* * *
For the author's responses below, although my suggestions have not completely been reflected to the revised manuscript, I understand that the author's thought and agree with the responses. I understand that one of the main scopes of this manuscript is describing the two-way nesting including multi domains / multi levels as a grid refinement method for ICON.

> *(my comment) L388: The multi-nesting using the recursive approach sounds interesting, however, the rest of this paper does not present any results from the multi-nesting experiments. Section 2.4 should be shortened and moved to Section 4 as future work, or*

*should be moved to Appendix.*

(author's response) In Section 2.4, we pointed out more clearly that multi-domain same-level and multi-level nesting is not an envisaged feature for the future, but that this feature is already implemented and used by the ICON community. What is, however, missing so far is a comprehensive technical description of this feature, which is one focus of this paper. An example of multi-domain same-level nesting is already given as part of the baroclinic wave tests in Sect. 3.1 (see Fig. 4e,f). Regarding multi-domain multi-level nesting we added a recent reference, where mountain-wave induced PSCs are simulated using a global ICON(-ART) configuration with three successively nested domains (Weimer et al.,2021).

*(my comment) L610: I suggest that the authors write down possible future works or plans of this study in the end of Section 4. Propagating the impacts of the two-way nesting downstream is the interesting and gives important implication. This finding implies the possibility of another nested domain upstream from the EU region for improving medium-range forecast over the EU region. This point could be a future prospect of this study in addition to multi-ways nesting described in 2.4.*

(author's response) We feel that discussing upcoming upgrades of our operational system would be beyond the scope of this paper, the primary purpose of which is to document the nesting implementation in ICON. Actually, we are preparing a horizontal resolution upgrade from 40/20 km to 26/13 km in the ensemble part of our system, and a vertical resolution upgrade from 90 to 120 levels in the deterministic as well as the ensemble part. We also considered extending the nested domain over the Atlantic towards the American coast, but the cost-benefit ratio of this change turned out to be not as good as for the general resolution increase. This is consistent with the widespread experience that the benefit of higher model resolution comes to a large extent from better resolving the orography, implying that the impact is more pronounced over land than over oceans.

---

## Author Response (AR2)

**GMD-2022-120:** Response to final comments (by editor)

*L510: "we note that the baroclinic instability of the real atmosphere is much smaller than in the initial state of the JW test...": "instability is smaller" appears to be grammatically strange. For an example, "we note that the disturbance growth due to the baroclinic instability in the real atmosphere is much weaker than in the initial state of the JW test..." could be an alternative.*

We clarified the formulation, now stating that the baroclinicity of the real atmosphere is weaker than in the JW test.

*L563: "On the other hand, this test has shown the ability of the child-to-parent feedback mechanism to have a small but noticeable positive impact on the quality of the nest interface conditions.": Interpreting "positive impact on the quality" corresponds to representing leeward propagation in the R2B11 parent domain more consistently with the R2B12 reference run, than that in the R2B11 without nesting. If this is correct, it could be formulated more explicitly around L560.*

We clarified the formulation in the preceding paragraph, emphasizing that a shorter feedback time scale improves the quality and consistency of the vertical nest interface condition, which in turn reduces the difference to the non-nested high-resolution reference run.

In addition, we fixed a few typos that we noticed during the final proofreading.

*Figure revisions:*

Comments by the editor:

We increased the font size of the labels in Fig. 4 and removed the y-axis tick labels from the right panels. In addition, we removed the x-axis tick labels from Fig. 7a. For Fig. 10 and 11, we are sorry that we were not able to increase the label font sizes due to the lack of access to the visualization software. However, we accomplished the requested revisions for Fig. 12, 13 and 14. For Fig. 14, we discarded the four uppermost pressure levels, which do not contain significant information, but decided to retain labels for each pressure level because these are not equidistant (i.e. the WMO standard pressure levels), implying that thinning the labels would reduce the accuracy of the information content.

Moreover, following a comment by Reviewer 2, we added boxes showing the boundaries of the nested domain to Figs. 13 and 15.